# *Drosophila* STING protein has a role in lipid metabolism

Katarina Akhmetova, Maxim Balasov, Igor Chesnokov*

Department of Biochemistry and Molecular Genetics, University of Alabama at Birmingham, School of Medicine, Birmingham, United States

**Abstract** Stimulator of interferon genes (STING) plays an important role in innate immunity by controlling type I interferon response against invaded pathogens. In this work, we describe a previously unknown role of STING in lipid metabolism in *Drosophila*. Flies with *STING* deletion are sensitive to starvation and oxidative stress, have reduced lipid storage, and downregulated expression of lipid metabolism genes. We found that *Drosophila* STING interacts with lipid synthesizing enzymes acetyl-CoA carboxylase (ACC) and fatty acid synthase (FASN). ACC and FASN also interact with each other, indicating that all three proteins may be components of a large multi-enzyme complex. The deletion of *Drosophila STING* leads to disturbed ACC localization and decreased FASN enzyme activity. Together, our results demonstrate a previously undescribed role of STING in lipid metabolism in *Drosophila*.

## Introduction

STimulator of INterferon Genes (STING) is an endoplasmic reticulum (ER)-associated transmembrane protein that plays an important role in innate immune response by controlling the transcription of many host defense genes (*Ishikawa and Barber, 2008*; *Ishikawa et al., 2009*; *Sun et al., 2009*; *Tanaka and Chen, 2012*; *Zhong et al., 2008*). The presence of foreign DNA in the cytoplasm signals a danger for the cell. This DNA is recognized by specialized enzyme, the cyclic GMP-AMP synthase (cGAS), which generates cyclic dinucleotide (CDN) signaling molecules (*Diner et al., 2013Li et al., 2013*; *Gao et al., 2013*; *Sun et al., 2013*). CDNs bind to STING activating it (*Wu et al., 2013*; *Burdette et al., 2011*), and the following signaling cascade results in NF-κB- and IRF3-dependent expression of immune response molecules such as type I interferons (IFNs) and pro-inflammatory cytokines (*Sun et al., 2009*; *Ishikawa et al., 2009*; *Liu et al., 2015a*). Bacteria that invade the cell are also known to produce CDNs that directly activate STING pathway (*Sauer et al., 2011*; *Woodward et al., 2010*; *Danilchanka and Mekalanos, 2013*). Additionally, DNA that has leaked from the damaged nuclei or mitochondria can also activate STING signaling and inflammatory response, which, if excessive or unchecked, might lead to the development of autoimmune diseases such as systemic lupus erythematosus or rheumatoid arthritis (*Ahn et al., 2012*; *Kawane et al., 2006*; *Jeremiah et al., 2014*; *Wang et al., 2015*).

STING homologs are present in almost all animal phyla (*Wu et al., 2014*; *Margolis et al., 2017*; *Kranzusch et al., 2015*). This protein has been extensively studied in mammalian immune response; however, the role of STING in the innate immunity of insects has been just recently identified (*Hua et al., 2018*; *Goto et al., 2018*; *Liu et al., 2018*; *Martin et al., 2018*). Fruit fly *D. melanogaster STING* homolog is encoded by the *CG1667* gene, which we hereafter refer to as *dSTING*. dSTING displays anti-viral and anti-bacterial effects that however are not all encompassing. Particularly, it has been shown that dSTING-deficient flies are more susceptible to *Listeria* infection due to the decreased expression of antimicrobial peptides (AMPs) – small positively charged proteins that possess antimicrobial properties against a variety of microorganisms (*Martin et al., 2018*). However, no effect has been observed during *Escherichia coli* or *Micrococcus luteus* infections (*Goto et al., 2018*). dSTING

*For correspondence:
ichesnokov@uab.edu

Competing interest: The authors declare that no competing interests exist.

has been shown to attenuate Zika virus infection in fly brains (*Liu et al., 2018*) and participate in the control of infection by two picorna-like viruses (DCV and CrPV) but not two other RNA viruses FHV and SINV or dsDNA virus IIV6 (*Goto et al., 2018*; *Martin et al., 2018*). All these effects are linked to the activation of NF-κB transcription factor Relish (*Kleino and Silverman, 2014*).

Immune system is tightly linked with metabolic regulation in all animals, and proper re-distribution of the energy is crucial during immune challenges (*Odegaard and Chawla, 2013*; *Alwarawrah et al., 2018*; *Lee and Lee, 2018*). In both flies and humans, excessive immune response can lead to a dysregulation of metabolic stores. Conversely, the loss of metabolic homeostasis can result in weakening of the immune system. The mechanistic links between these two important systems are integrated in *Drosophila* fat body (*Arrese and Soulages, 2010*; *Buchon et al., 2014*). Similarly to mammalian liver and adipose tissue, insect fat body stores excess nutrients and mobilizes them during metabolic shifts. The fat body also serves as a major immune organ by producing AMPs during infection. There is an evidence that the fat body is able to switch its transcriptional status from 'anabolic' to 'immune' program (*Clark et al., 2013*). The main fat body components are lipids, with triacylglycerols (TAGs) constituting approximately 90 % of the stored lipids (*Canavoso et al., 2001*). Even though most of the TAGs stored in fat body comes from the dietary fat originating from the gut during feeding, de novo lipid synthesis in the fat body also significantly contributes to the pool of storage lipids (*Heier and Kühnlein, 2018*; *Wicker-Thomas et al., 2015*; *Parvy et al., 2012*; *Garrido et al., 2015*).

Maintaining lipid homeostasis is crucial for all organisms. Dysregulation of lipid metabolism leads to a variety of metabolic disorders such as obesity, insulin resistance and diabetes. Despite the difference in physiology, most of the enzymes involved in metabolism, including lipid metabolism, are evolutionarily and functionally conserved between *Drosophila* and mammals (*Lehmann, 2018*; *Toprak et al., 2020*). Major signaling pathways involved in metabolic control, such as insulin system, TOR, steroid hormones, FOXO, and many others, are present in fruit flies (*Brogiolo et al., 2001*; *Oldham et al., 2000*; *Jünger et al., 2003*; *King-Jones and Thummel, 2005*). Therefore, it is not surprising that *Drosophila* has become a popular model system for studying metabolism and metabolic diseases (*Teleman, 2009*; *Owusu-Ansah and Perrimon, 2014*; *Musselman and Kühnlein, 2018*; *Baker and Thummel, 2007*; *Liu and Huang, 2013*; *Graham and Pick, 2017*; *Diop and Bodmer, 2015*). With the availability of powerful genetic tools, *Drosophila* has all the advantages to identify new players and fill in the gaps in our understanding of the intricacies of metabolic networks.

In this work, we describe a novel function of dSTING in lipid metabolism. We report that flies with a deletion of *dSTING* are sensitive to the starvation and oxidative stress. Detailed analysis reveals that *dSTING* deletion results in a significant decrease in the main storage metabolites, such as TAG, trehalose, and glycogen. We identified two fatty-acid biosynthesis enzymes – acetyl-CoA carboxylase (ACC) and fatty acid synthase (FASN) – as the interacting partners for dSTING. Moreover, we also found that FASN and ACC interacted with each other, indicating that all three proteins might be components of a large complex. Importantly, *dSTING* deletion leads to the decreased FASN activity and defects in ACC cellular localization suggesting a direct role of dSTING in lipid metabolism of fruit flies.

## Results

### *Drosophila STING* mutants are sensitive to starvation and oxidative stress

Previously, we described a large genomic deletion that included *orc6* gene and the neighboring *CG1667* (*dSTING*) gene, which at that time was not characterized (*Balasov et al., 2009*). To create a specific *dSTING* mutation, we used the method of *P*-element imprecise excision. A *P*-element-based transposon P{EPgy2} Sting^EY0649^, located 353 base pairs upstream of the *dSTING* start codon, was excised by Δ2–3 transposase. *dSTINGΔ* allele contained deletion of 589 base pairs including start codon, first exon, and part of the second exon (*Figure 1A*). Homozygous *dSTINGΔ* mutant flies are viable with no obvious observable phenotype.

The role of dSTING in anti-viral and anti-bacterial defense in *Drosophila* has been established recently (*Martin et al., 2018*; *Goto et al., 2018*; *Liu et al., 2018*). Since the changes in immune response are often accompanied by a dysregulation of metabolic homeostasis and vice versa (*Zmora et al., 2017*; *Odegaard and Chawla, 2013*; *Alwarawrah et al., 2018*), we analyzed *dSTINGΔ* mutant flies for the defects in metabolism. A response to the metabolic stress is a good indicator of defects in

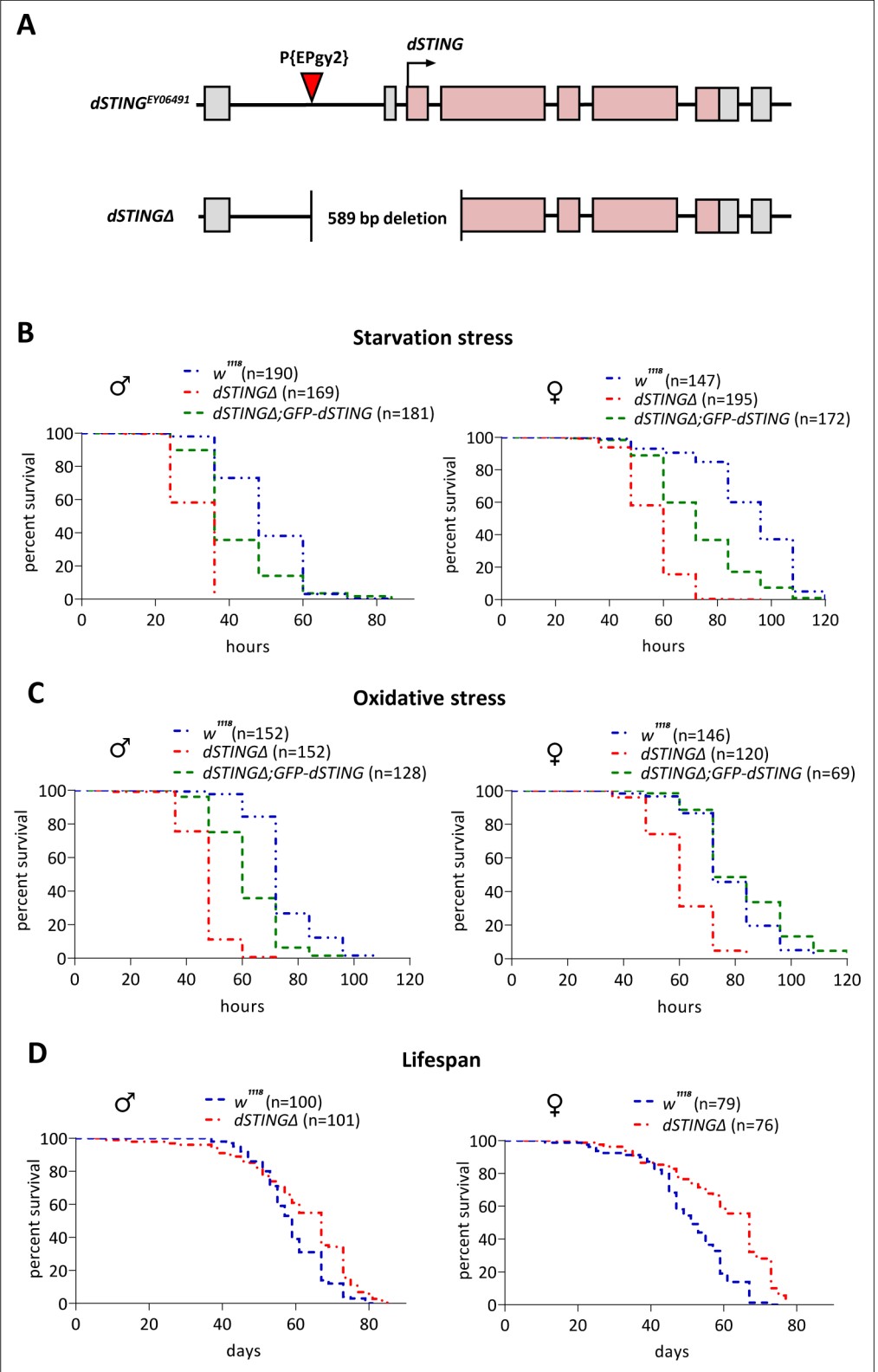

**Figure 1.** *Drosophila STING* mutants are susceptible to starvation and oxidative stress but have normal life span. (**A**) Generation of the *Drosophila STING* deletion mutant. *dSTING* deletion mutant was generated by imprecise excision of P-element *P{EPgy2}STING^EY06491^*. *dSTINGΔ* allele contains a deletion of 589 base pairs including start codon, first exon, and part of the second exon of *dSTING*. Exons are shown as pink-colored rectangles. The position of the P-element insertion is indicated by the red triangle. (**B**) Starvation stress resistance of males and

*Figure 1 continued on next page*

*Figure 1 continued*

females. Five-day-old flies were kept on PBS only and percentages of surviving flies were counted every 12 hr. (**C**) Oxidative stress resistance of males and females. Five-day-old flies were kept on food supplemented with 5 % hydrogen peroxide and percentages of surviving flies were counted every 12 hr. (**D**) Lifespan of males and females. Flies were kept on regular food and percentages of surviving flies were counted. (**B–D**) Genotypes used were: control flies – *w1118*, flies with *dSTING* deletion – *dSTINGΔ*, genetic rescue – *dSTINGΔ;GFP-dSTING*. Percentages of surviving flies at each time point are shown. The number of flies analyzed is shown in chart legend for each genotype. Log-rank test yielded p<0.001 for all pairwise comparisons except for (**C**): *w1118* vs *dSTINGΔ;GFP-dSTING* females under oxidative stress showed no statistical significance (p=0.121).

The online version of this article includes the following figure supplement(s) for figure 1:

**Source data 1.** Source file for survival curves.

**Figure supplement 1.** GFP-tagged *Drosophila STING* expression in different tissues.

**Figure supplement 1—source data 1.** Source file for GFP-dSTING expression.

**Figure supplement 1—source data 2.** Source file for GFP-dSTING expression (adult tissues).

**Figure supplement 1—source data 3.** Source file for GFP-dSTING expression (larval tissues).

**Figure supplement 2.** *Drosophila STING* mutant larvae are susceptible to starvation and oxidative stress.

**Figure supplement 3.** *Drosophila STING* mutants are susceptible to starvation and oxidative stress and have decreased TAG and glycogen levels under axenic condition.

metabolism; therefore, we subjected flies to a starvation stress and an oxidative stress. After the eclosion, flies were kept on regular food for 5 days and then transferred to vials containing wet Whatman paper (starvation stress) or to vials containing regular food supplemented with 5 % hydrogen peroxide (oxidative stress). The percentages of surviving flies were counted every 12 hr. We found that *dSTINGΔ* mutant flies were sensitive to both starvation and oxidative stress as compared to the control flies (*Figure 1B,C*).

To confirm that the observed phenotypes are not due to off-target effects, we designed fly strain containing GFP-tagged wild-type *dSTING* (under the native *dSTING* promoter) on the *dSTINGΔ* deletion background. The level of *dSTING* expression in *dSTINGΔ;GFP-dSTING* flies was the same as in control flies (*Figure 1—figure supplement 1A*). We also looked at the expression pattern of *GFP-dSTING* across adult and larval tissues. The highest level of the expression was observed in the digestive tract in both adults and larvae. *GFP-dSTING* was also expressed at the high level in the larval fat body and adult abdominal carcasses which are enriched in fat body cells (*Zhao and Karpac, 2017*; *Molaei et al., 2019*; *Lőrincz et al., 2017*; *Figure 1—figure supplement 1B,D*). Our results were consistent with the modENCODE Tissue Expression Data for *dSTING* (*Brown et al., 2014*; *Figure 1—figure supplement 1C,E*).

Importantly, the expression of *GFP-dSTING* partially or entirely rescued the sensitivity of *dSTINGΔ* deletion flies to both starvation and oxidative stress (*Figure 1B,C*), suggesting that the observed phenotypes are caused by dSTING deficiency. The larvae carrying *dSTINGΔ* deletion were also more susceptible to both types of stress (*Figure 1—figure supplement 2*). The deletion of *dSTING* had no effect on the total lifespan of fed flies in both males and females. Moreover, the age-related mortality was slightly reduced, especially for the females (*Figure 1D*).

It is possible that the increased sensitivity to starvation and oxidative stress that we observed in *dSTINGΔ* flies is caused by a lowered defense against commensal or pathogenic bacteria in the absence of *dSTING*. To test this hypothesis, we generated axenic, or germ-free, flies. We found that under axenic condition *dSTINGΔ* mutants exhibited the same response to the starvation and oxidative stress as *dSTINGΔ* non-axenic flies (*Figure 1—figure supplement 3A,B*). This suggests that diminished immune response against bacteria is not likely to be the cause of the observed phenotypes.

Collectively, our data suggest that the deletion of *Drosophila STING* results in an increased susceptibility of flies to starvation and to oxidative stress.

## *Drosophila STING* mutants have decreased storage metabolites

The ability of an organism to store nutrients when they are abundant is crucial for its survival during periods of food shortage. Triacylglycerols (TAGs), glycogen, and trehalose are the major metabolites for a carbon storage in *Drosophila*. Dietary glucose absorbed from the gut is quickly converted to

trehalose, which is a main hemolymph sugar in insects. Glycogen is another form of a carbohydrate storage that accumulates in the fat body and muscles. Finally, most energy reserves in insects are in the form of lipids, particularly TAGs that are stored in the lipid droplets of the fat body (*Canavoso et al., 2001*).

We measured storage metabolite levels along with glucose level in fed or 24 hr starved adult males. Under fed conditions, TAG level was decreased twofold in *dSTINGΔ* mutants compared with the control flies. Under starved conditions, TAG level dropped dramatically to about 1/8 of the level in the control flies (*Figure 2A*). Glycogen and trehalose levels were also significantly decreased in *dSTINGΔ* mutants in both fed and starved flies (*Figure 2B,C*). Interestingly, glucose level was increased under fed condition (*Figure 2D*, fed), suggesting that *dSTINGΔ* mutant flies might have either a decreased incorporation of ingested glucose into trehalose or glycogen, or an increased breakdown of these storage molecules. Nevertheless, when flies were starved for 24 hr, glucose level in *dSTINGΔ* mutants dropped and was twofold lower than in control flies (*Figure 2D*, starved). In the fed flies, the expression of GFP-tagged *dSTING* partially rescued the mutant phenotypes for all measured metabolites (*Figure 2A–D*, fed). However, under starved condition, the rescue was observed only for TAG level (*Figure 2A–D*, starved). Importantly, axenic *dSTINGΔ* mutants still showed decreased TAG and glycogen levels as compared to the axenic control flies (*Figure 1—figure supplement 3C,D*), suggesting that the lowered storage metabolite levels are not due to the diminished immune response against bacteria.

Two RNAi screens for obesity and anti-obesity genes in *Drosophila* did not reveal any significant changes in TAG level in dSTING-deficient flies (*Pospisilik et al., 2010*; *Baumbach et al., 2014*). The potential discrepancy with our data might be explained by the fact that in both mentioned studies RNAi was induced only 2–8 days after the eclosion, whereas in our study, dSTING was absent from the very beginning of the development.

One of the possible explanations for the decreased storage metabolites might be a decrease in food consumption. To test this possibility, we used capillary feeder (CAFE) assay (*Ja et al., 2007*), which showed that it was not the case, and *dSTINGΔ* mutant flies consumed food at the same rate as control flies (*Figure 2—figure supplement 1A*).

Also, a compromised gut barrier function could potentially lead to a decreased nutrient absorption and susceptibility to starvation stress. To assess an intestinal permeability in vivo we performed 'smurf' assay (*Rera et al., 2012*). Flies were fed blue dye and checked for the presence of the dye outside of the digestive tract. 'Smurf' assay did not reveal any loss of gut wall integrity in *dSTINGΔ* mutants (*Figure 2—figure supplement 1B*).

Together, these data indicate that a deletion of *dSTING* results in the decreased levels of storage molecules, with the effect most pronounced for TAGs.

## Lipid metabolism is impaired in *Drosophila STING* mutants

Among measured metabolites, the effect of *dSTING* mutation on TAG level was the most pronounced. Moreover, the expression of *GFP-dSTING* on *dSTINGΔ* mutant background partially rescued TAG levels under both fed and starved conditions (*Figure 2A*). Therefore, we decided to take a closer look at the lipid metabolism in the absence of dSTING. In insects, TAGs are stored mainly in the fat body and midgut in the form of cytoplasmic lipid droplets. To visualize the lipid stores, we stained fat bodies and midguts of adult flies with Nile Red dye that selectively labels lipids within the cells (*Figure 2E,F*). Fat bodies of *dSTINGΔ* mutant flies contained significantly fewer lipids as compared to the control flies (*Figure 2E,E'*). The expression of *GFP-dSTING* rescued this phenotype. Staining with the another lipid-specific dye, LipidTox, showed similar results (*Figure 2—figure supplement 2*). Interestingly, lipid droplet content in midguts was not decreased in *dSTINGΔ* mutants (*Figure 2F,F'*), indicating that only the fat body lipid storage was affected.

Next, we performed *dSTING* RNAi using female fat-body-specific *yolk-GAL4* driver. Flies with reduced *dSTING* expression specifically in fat body were more susceptible to the starvation stress and oxidative stress, and had reduced TAG and glycogen levels (*Figure 2—figure supplement 3*), highlighting the role of dSTING in fat body functions.

To gain an insight into gene expression changes in the absence of *dSTING*, we performed microarray analysis of *dSTINGΔ* mutant and control flies under the fed and 24 hr starved conditions. Under fed conditions, microarray analysis revealed a significant change in 672 transcripts (more than

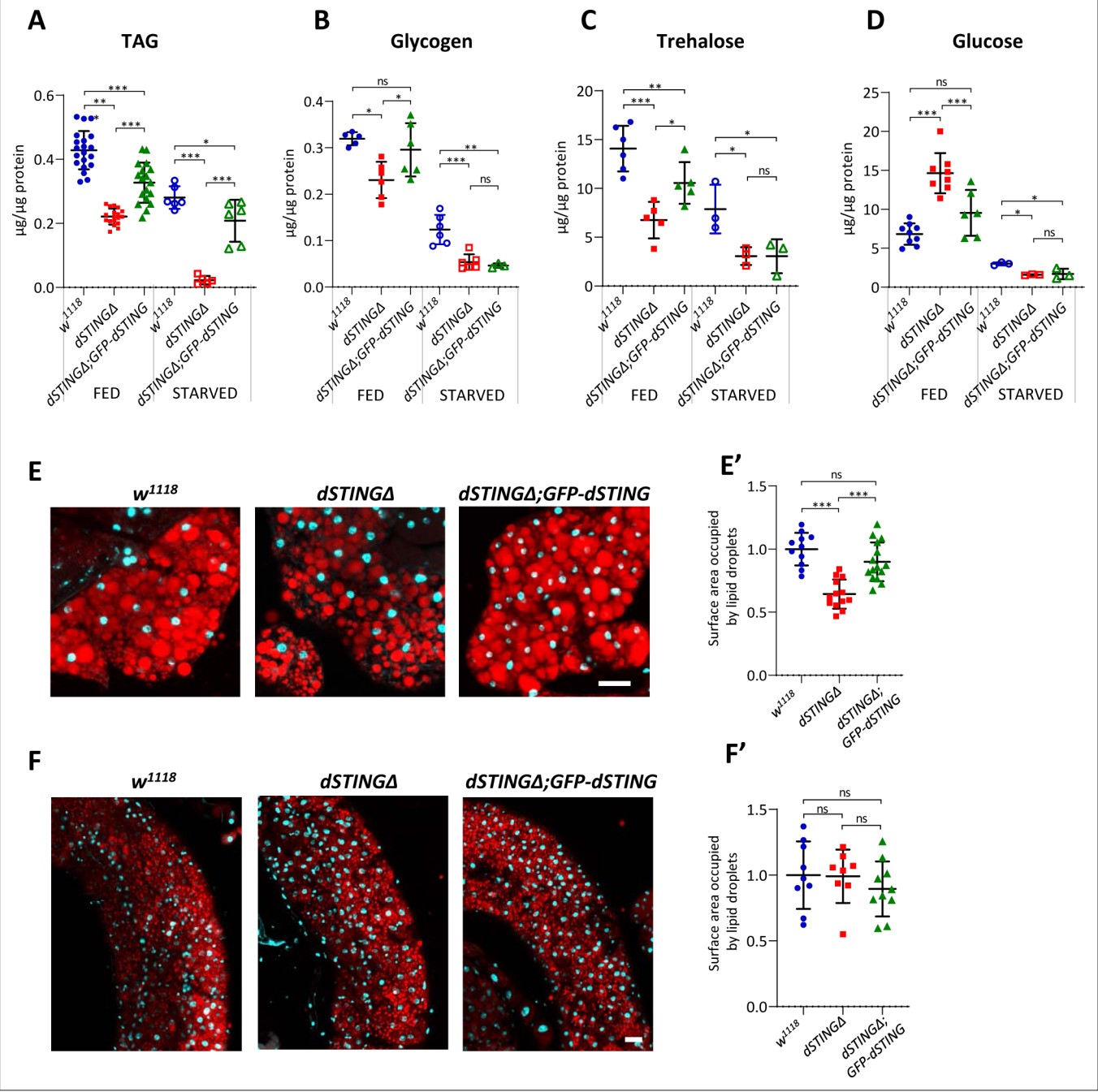

**Figure 2.** Storage metabolites are significantly decreased in *Drosophila STING* mutants. (**A–D**) Metabolites levels in fed or 24 hr starved 5-day-old males. TAG (**A**) and glycogen (**B**) levels were measured in the total body. Trehalose (**C**) and glucose (**D**) levels were measured in the hemolymph. Levels of metabolites are shown per µg of total protein. Data are represented as mean ± SD. One-way ANOVA with Tukey's post hoc test. *p<0.05, **p<0.01, ***p<0.001, ns indicates statistically non-significant. (**E, F**) Staining of male adult tissues for lipid content. Fat bodies (**E**) or midguts (**F**) were stained with Nile Red (red) that labels lipid droplets. Nuclei were stained with DAPI (blue). Scale bar 20 µm. (**E', F'**) Quantification of surface area occupied by lipid droplets in fat bodies (**E'**) and midguts (**F'**). Values are normalized to the wild type (*w1118*). Data are represented as mean ± SD. One-way ANOVA with Tukey's post hoc test. ***p<0.001, ns indicates statistically non-significant. Genotypes used were: control flies – *w1118*, flies with *dSTING* deletion – *dSTINGΔ,* genetic rescue – *dSTINGΔ;GFP-dSTING*.

The online version of this article includes the following figure supplement(s) for figure 2:

**Source data 1.** Source file for metabolite levels.

**Figure supplement 1.** Food ingestion is not compromised in *Drosophila STING* mutant flies.

**Figure supplement 2.** *Drosophila STING* mutants have reduced TAG storage in fat body.

*Figure 2 continued on next page*

*Figure 2 continued*

**Figure supplement 3.** Fat body-specific RNAi of *Drosophila STING* results in increased sensitivity to starvation and oxidative stress and decreased TAG and glycogen levels.

1.4 fold change), with 381 transcripts expressed at reduced levels and 291 transcripts at elevated levels. Under starved conditions, the expression of 1452 genes was altered in *dSTINGΔ* mutants, with 797 downregulated and 655 upregulated genes (***Supplementary file 1***).

Principal component analysis (PCA) is a common method for the analysis of gene expression data, providing an information on the overall structure of the analyzed dataset (***Lever et al., 2017***). PCA plot for our microarray data showed that the sample groups separated along the PC1 axis (which explained 29 % of all variance in the experiment), with the greatest separation between the control fed and mutant starved groups (***Figure 3A***). Interestingly, the *dSTINGΔ* fed and the control starved

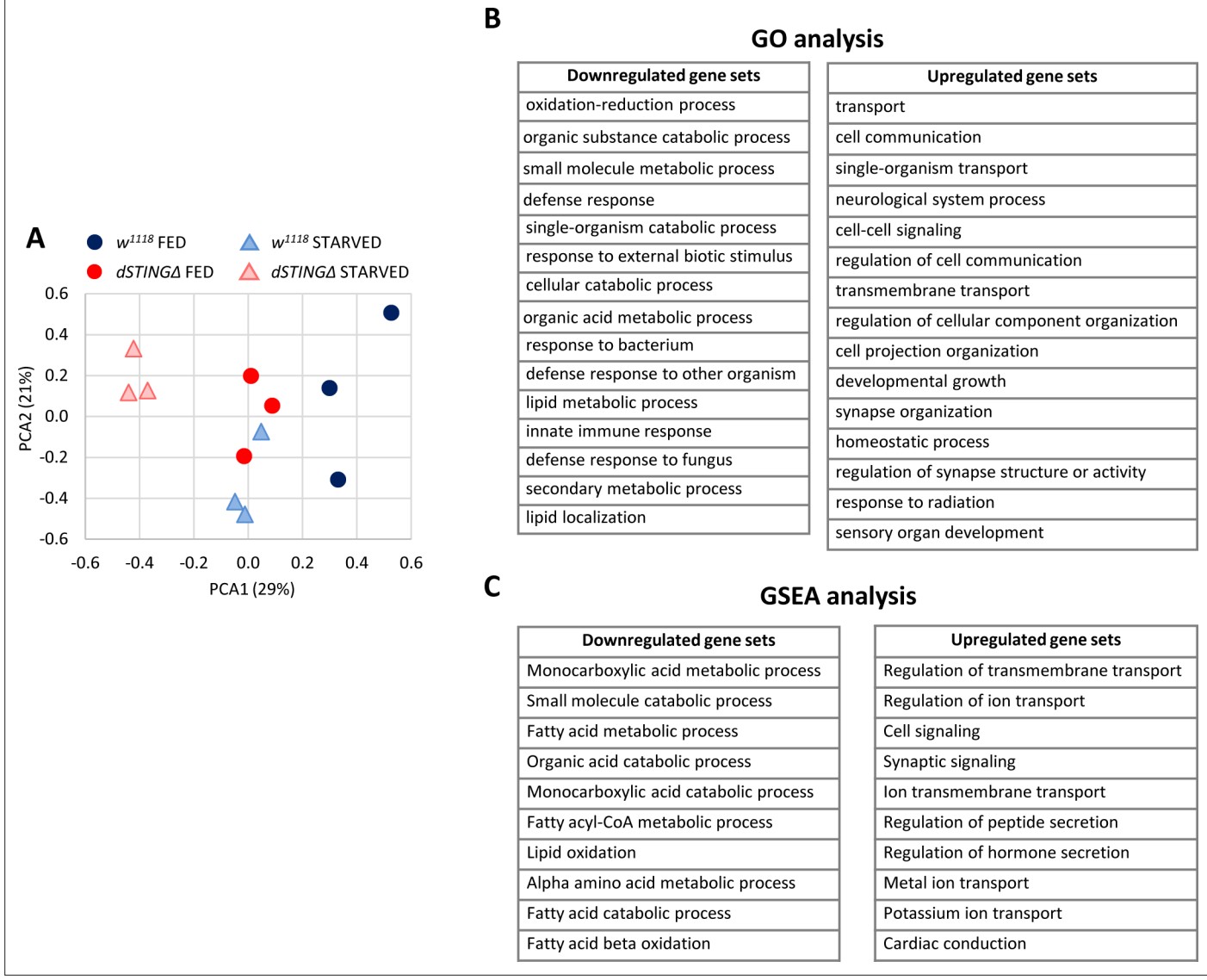

**Figure 3.** Lipid metabolism genes are downregulated in *Drosophila STING* mutants. Fed or 24 hr starved 5-day-old adult males (*dSTINGΔ* mutants or *w1118* as a control) were subjected to microarray analysis. (**A**) Principal component analysis (PCA) of microarray data. PCA scores plot showing variances in gene expression profiles between groups is shown. Each sample is shown as a single point (n = 3 per genotype). (**B**) Gene ontology (GO) analysis of microarray data. *dSTINGΔ* mutants and control *w1118* under fed conditions were compared. Downregulated and upregulated top scoring gene sets are shown. (**C**) Gene Set Enrichment Analysis (GSEA) of microarray data. *dSTINGΔ* mutants and control *w1118* under fed conditions were compared. Downregulated and upregulated top scoring gene sets are shown.

groups clustered together along PC1 axis, indicating that the *dSTING* knockout and the starvation induced similar changes in gene expression.

In agreement with the previous report, *dSTINGΔ* mutants are characterized by the downregulation of immune response genes, including AMPs (*Mtk*, *Drs*, *AttD*, *DptB*, *BomS1*), peptidoglycan recognition proteins (PGRPs, such as *PGRP-SD* and *PGRP-SA*), and serpins (*Spn53F*, *Spn42De*) (*Supplementary file 1*; *Martin et al., 2018*). These results are expected since *STING* was initially discovered in fruit flies and silkworm as an immune response gene (*Martin et al., 2018*; *Hua et al., 2018*; *Goto et al., 2018*). To gain more insight into the biological processes that are altered in the absence of dSTING, we looked at the gene set enrichment in *dSTINGΔ* mutants under fed conditions. Based on the Gene Ontology (GO) analysis, metabolic processes and immune response genes were downregulated in *dSTINGΔ* mutants (*Figure 3B*, downregulated gene sets). Upregulated genes were enriched with GO classifications related to cell signaling (e.g. transport, cell communication and synapse organization) (*Figure 3B*, upregulated gene sets).

GO analysis requires a discrete list of genes (downregulated and upregulated in our case). Gene Set Enrichment Analysis (GSEA), on the other hand, uses all microarray data points; therefore, it is expected to be more sensitive since it can identify gene sets comprising many members that are undergoing subtle changes in the expression (*Subramanian et al., 2005*). We analyzed *dSTINGΔ* mutants versus control flies under fed conditions using GSEA approach and found that the metabolism of lipids, particularly fatty acids, was among top scoring gene sets downregulated in *dSTINGΔ* mutants (*Figure 3C*, downregulated gene sets). As for upregulated gene sets, GSEA data were similar to GO analysis data (*Figure 3C*, upregulated gene sets).

Together, we found that *Drosophila STINGΔ* mutants have defects in lipid metabolism manifested in the decreased lipid storage in fat body and the decreased expression of lipid metabolism genes.

## *Drosophila STING* protein interacts with ACC and FASN

In mammals, STING is an adaptor molecule that activates downstream signaling through protein–protein interactions. To look for possible interaction partners for *Drosophila* STING that could explain its effect on lipid metabolism, we performed the immunoprecipitation from fat bodies of larvae expressing *GFP-dSTING* using anti-GFP antibody. Immunoprecipitated material was separated by SDS–PAGE, and the most prominent bands were subjected to mass spectrometry analysis. Several proteins with a high score were identified, including FASN 1 and 2 (CG3523 and CG3524, respectively), ACC (CG11198), and dSTING itself (CG1667) (*Supplementary file 2*).

ACC and FASN are two important enzymes of the de novo lipid biosynthesis pathway (*López-Lara and Soto, 2019*; *Wakil and Abu-Elheiga, 2009*). ACC catalyzes the formation of malonyl-CoA from acetyl-CoA, the first committed step of fatty acid synthesis. The next step is performed by FASN, which uses malonyl-CoA and acetyl-CoA to synthesize palmitic fatty acid. Palmitate might undergo a separate elongation and/or unsaturation by specialized enzymes to yield other fatty acid molecules. A series of reactions then add the fatty acids to a glycerol backbone to form triacylglycerol (TAG), the main energy storage molecule.

We confirmed the mass spec results by performing the immunoprecipitation from the abdomens of adult flies expressing *GFP-dSTING* in fat body. Both ACC and FASN co-immunoprecipitated with GFP-dSTING (*Figure 4A,B*). Interestingly, we found that ACC and FASN interacted with each other as showed by the reciprocal immunoprecipitation experiments (*Figure 4C,D*). We observed this interaction not only in control flies but also in *dSTINGΔ* mutants flies with or without the expression of *GFP-dSTING*.

Together, our data indicate that dSTING, ACC, and FASN interact with one another, suggesting that they might be components of a multi-protein complex involved in fatty acid synthesis.

## FASN activity is decreased in *Drosophila STING* mutants

Since dSTING was found to interact with ACC and FASN, we asked if *dSTING* deletion might result in changes in these enzymes' activities. We measured ACC and FASN activity in adult flies. ACC protein level and activity were not significantly changed in *dSTINGΔ* mutants flies (*Figure 5A,A'*). However, FASN activity was almost two times lower in *dSTINGΔ* mutants as compared to the control flies (*Figure 5B*), but the protein level was unchanged (*Figure 5B'*). Importantly, the expression of GFP-tagged dSTING on *dSTINGΔ* mutant background restored FASN activity to the control level (*Figure 5B*).

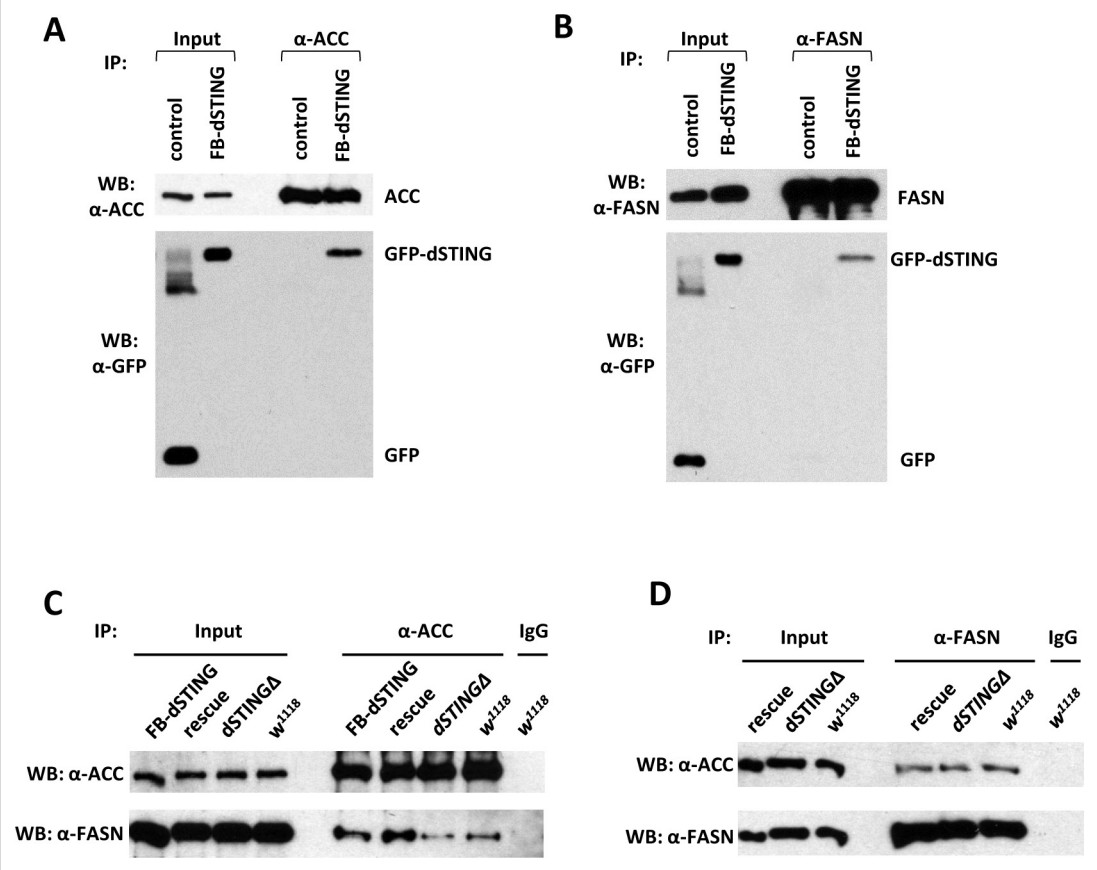

**Figure 4.** *Drosophila* STING protein interacts with acetyl-CoA carboxylase (ACC) and fatty acid synthase (FASN). (**A, B**) dSTING interacts with ACC and FASN. ACC (**A**) or FASN (**B**) were immunoprecipitated from abdomens of adult flies using corresponding antibody. 'Control' – *w¹¹¹⁸*, "FB-dSTING" – *cg-GAL4/GFP-dSTING* (flies expressing *GFP-dSTING* in fat body). Recombinant GFP was added to the control reaction. (**C, D**) ACC and FASN interact with each other. ACC (**C**) or FASN (**D**) were immunoprecipitated from abdomens of adult flies using corresponding antibody. "Rescue" – *dSTINGΔ;GFP-dSTING*, "FB-dSTING" – *cg-GAL4/GFP-dSTING* (flies expressing *GFP-dSTING* in fat body).

The online version of this article includes the following figure supplement(s) for figure 4:

**Source data 1.** Source file for ACC, dSTING and FASN IP experiments.

**Source data 2.** Source file for *Figure 4A* (ACC and dSTING IP experiment).

**Source data 3.** Source file for *Figure 4B* (dSTING and FASN IP experiment).

**Source data 4.** Source file for *Figure 4C* (ACC and FASN IP experiment).

**Source data 5.** Source file for *Figure 4D* (ACC and FASN IP experiment).

ACC enzyme carboxylates acetyl-CoA resulting in the formation of malonyl-CoA, which then serves as a substrate for FASN in the synthesis of fatty acids. If ACC activity is unchanged and FASN activity is decreased, we should observe the accumulation of malonyl-CoA. Polar metabolite profiling of *dSTINGΔ* flies compared to control flies showed that indeed, malonyl-CoA level was significantly increased, whereas acetyl-CoA level remained unchanged in the mutants (*Figure 5C*, *Figure 5—figure supplement 1*).

## ACC localization is perturbed in the fat body of *dSTINGΔ* mutants

In mammals, STING is a transmembrane protein that localizes to the ER. To check whether this is also the case in *Drosophila*, we performed membrane fractionation, which showed that GFP tagged dSTING co-sedimented exclusively with the membrane fraction (*Figure 6—figure supplement 1A*). We also expressed *GFP-dSTING* in *Drosophila* S2 tissue culture cells and found that it mostly co-localized with the ER and to the lesser extent with the Golgi, but not with the cellular membrane (*Figure 6—figure supplement 1B*), in agreement with previous observation (*Goto et al., 2018*).

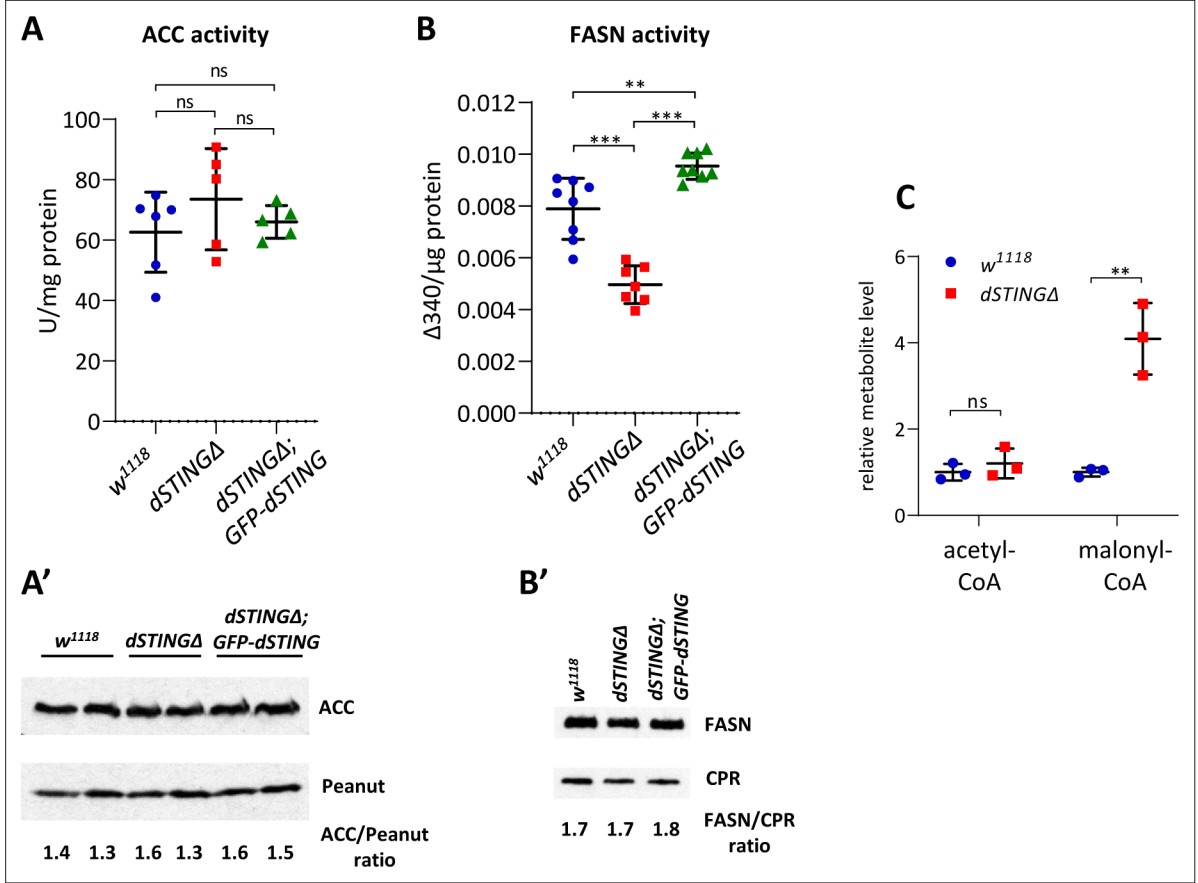

**Figure 5.** Fatty acid synthase activity is decreased in *Drosophila STING* mutants. (**A, B**) Enzyme activity assays. ACC activity (**A**) or FASN activity (**B**) was measured in the total body of adult flies and normalized to protein level. The number of experiments for each genotype is indicated. Data are represented as mean ± SD. One-way ANOVA with Tukey's post hoc test. **$p<0.01$, ***$p<0.001$, ns indicates statistically non-significant. (**A'**) ACC protein level in total fly extract. Peanut was used as a loading control. (**B'**) FASN protein level in total fly extract. CPR (NADPH-cytochrome P450 reductase) was used as a loading control. (**C**) Acetyl-CoA and malonyl-CoA levels in fly total body extracts. Values were normalized to wild type ($w^{1118}$). Data are represented as mean ± SD. Student's t-test, **$p<0.01$, ns indicates statistically non-significant.

The online version of this article includes the following source data and figure supplement(s) for figure 5:

**Source data 1.** Source file for ACC and FASN protein level experiment.

**Source data 2.** Source file for *Figure 5A'* (ACC protein level).

**Source data 3.** Source file for *Figure 5B'* (FASN protein level).

**Source data 4.** Source file for enzyme activity levels.

**Figure supplement 1.** Metabolomics analysis of *Drosophila STING* mutants.

**Figure supplement 1—source data 1.** Quantitation of polar metabolites in 5-days old adult males.

Next, we used adult flies expressing GFP-tagged *dSTING* under the native *dSTING* promoter to examine the localization of dSTING, ACC, and FASN in the fat body, main lipid synthesizing organ in *Drosophila*. The ER (as judged by the ER marker Calnexin) extended throughout fat body cells, with most prominent staining at the cell periphery and in the perinuclear region as was shown before (*Jacquemyn et al., 2020*; *Figure 6A, A'*). GFP-dSTING mainly co-localized with Calnexin at the cortex. Little or no signal was observed at the perinuclear region of fat body cells (*Figure 6A,A'*). Both ACC and FASN partially co-localized with GFP-dSTING at the cell periphery region of the ER (*Figure 6B, B', C and C'*).

We asked whether *dSTINGΔ* mutation affected localization of ACC or FASN in fly fat body. While ACC extended throughout wild-type cells, in *dSTINGΔ* mutant cells ACC concentrated in the perinuclear region with a minimal signal in the cell periphery (*Figure 7A*). The expression of *GFP-dSTING* on *dSTING* null background normalized ACC staining toward the wild-type distribution. Calnexin staining was not affected by the mutation. A closer examination of the perinuclear region of the fat body cells revealed

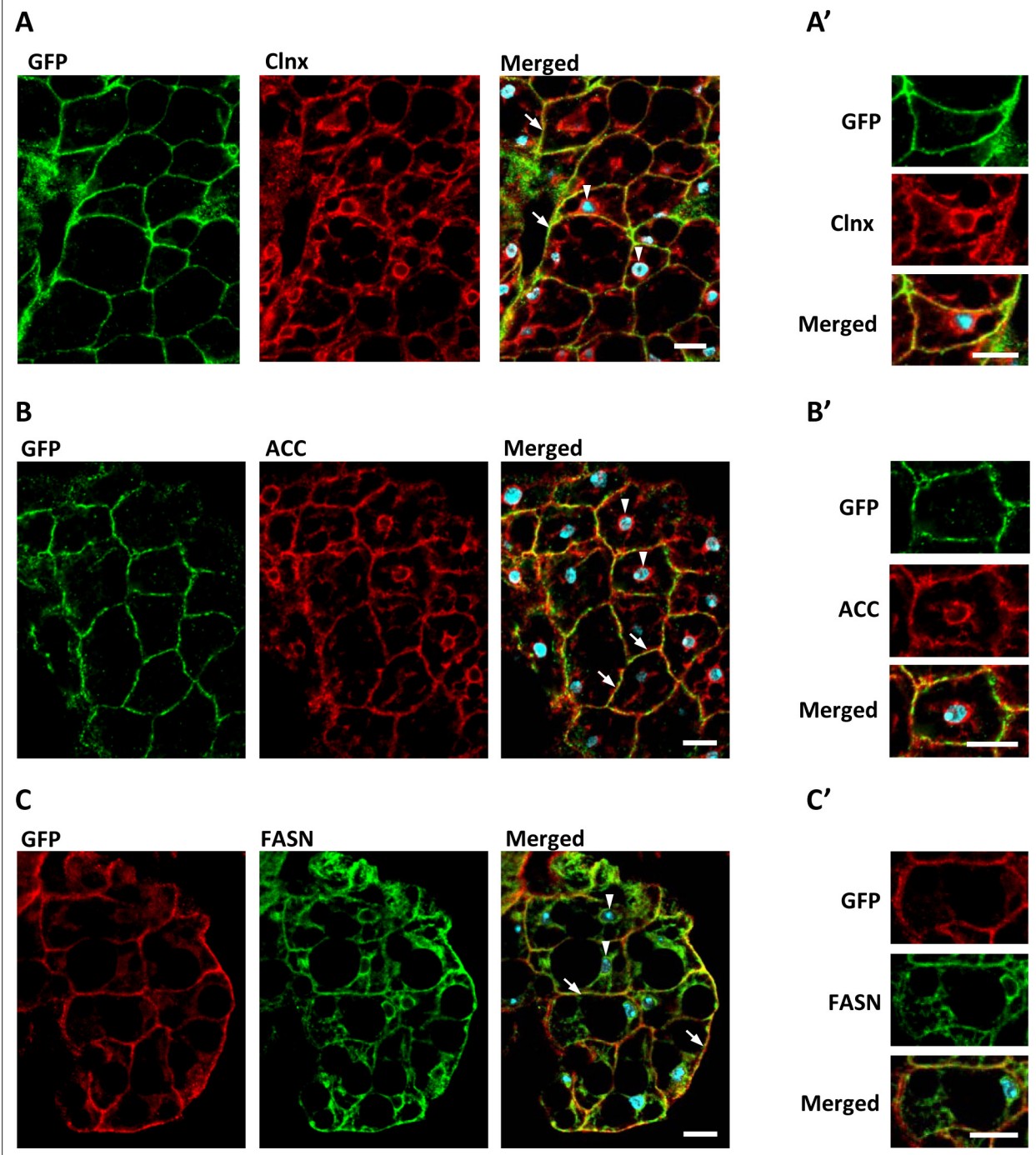

**Figure 6.** dSTING, ACC, and FASN co-localize in *Drosophila* fat body cells. Fat body of adult flies expressing GFP-tagged dSTING (genotype *dSTINGΔ;GFP-dSTING*) were stained for: (**A, A'**) GFP (green) and Calnexin (Clnx, red); (**B, B'**) GFP (green) and ACC (red); (**C, C'**) GFP (red) and FASN (green). Nuclei were stained with DAPI (blue). Arrows mark cortical region, arrowheads mark perinuclear region of fat body cells. Scale bar 10 μm. Higher magnification is shown at (**A', B', C'**).

The online version of this article includes the following figure supplement(s) for figure 6:

**Figure supplement 1.** *Drosophila* STING localizes at the endoplasmic reticulum (ER) membrane.

**Figure supplement 1—source data 1.** Source file for membrane fractionation experiment.

**Figure supplement 1—source data 2.** Source file for membrane fractionation experiment.

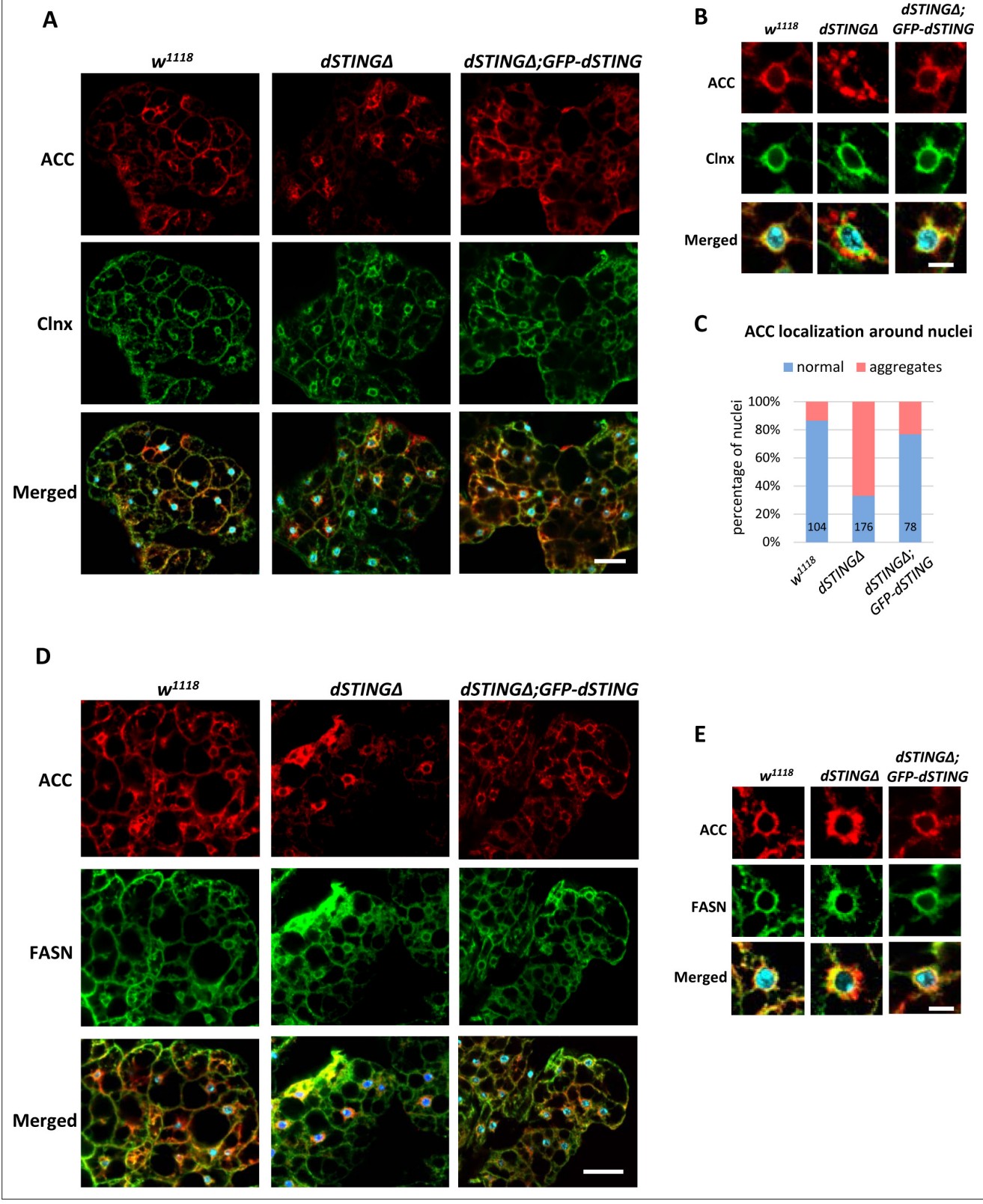

**Figure 7.** ACC localization is perturbed in *Drosophila STING* mutant fat body. (**A–C**) Adult fat bodies were stained with ACC (red), Calnexin (Clnx, green), and DAPI (blue). (**A**) ACC has decreased cortical localization in *dSTINGΔ* mutant fat body as compared to control (*w^1118^*) and 'rescue' (*dSTINGΔ;GFP-dSTING*) fly strains. Scale bar 20 μm. (**B**) ACC localization in the perinuclear region of fat body cells. Scale bar 5 μm. (**C**) Quantification of perinuclear ACC localization pattern. Numbers of nuclei analyzed are shown for each genotype. (**D–E**) Adult fat bodies were stained with ACC

*Figure 7 continued on next page*

*Figure 7 continued*

(red), FASN (green), and DAPI (blue). (**D**) FASN localization is not changed in *dSTINGΔ* mutant fat body as compared to control (*w1118*) and "rescue" (*dSTINGΔ;GFP-dSTING*) fly strains. Scale bar 20 µm. (**E**) FASN and ACC localization in the perinuclear region of fat body cells. Scale bar 5 µm.

that in *dSTINGΔ* mutants, ACC appeared disorganized and aggregated compared with the wild-type cells and cells expressing *GFP-dSTING* (*Figure 7B*). The quantifications showed that 67 % of nuclei had a perinuclear 'aggregated' ACC phenotype (*Figure 7C*). On the other hand, FASN maintained its localization pattern in *dSTINGΔ* mutant cells (*Figure 7D*), but partially co-localized with ACC 'aggregates' (*Figure 7E*), in agreement with the immunoprecipitation results (*Figure 4C,D*).

Thus, we conclude that the presence of dSTING in fat body cells is required for the proper ACC localization. In the absence of dSTING, ACC is no longer able to localize at the cell periphery and forms aggregated structures around fat body nucleus. FASN localization is mostly unchanged in *dSTINGΔ* mutants.

## Discussion

STING plays an important role in innate immunity of mammals, where activation of STING induces type I interferons (IFNs) production following the infection with intracellular pathogens (*Ishikawa and Barber, 2008*; *Ishikawa et al., 2009*; *Sun et al., 2009*; *Tanaka and Chen, 2012*; *Zhong et al., 2008*). However, recent studies showed that the core components of STING pathway evolved more than 600 million years ago, before the evolution of type I IFNs (*Wu et al., 2014*; *Margolis et al., 2017*; *Morehouse et al., 2020*). This raises the question regarding the ancestral functions of STING. In this study we found that STING protein is involved in lipid metabolism in *Drosophila*. The deletion of *Drosophila STING* (*dSTING*) gene rendered flies sensitive to the starvation and oxidative stress. These flies have reduced lipid storage and downregulated expression of lipid metabolism genes. We further showed that dSTING interacted with the lipid synthesizing enzymes ACC and FASN suggesting a possible regulatory role in the lipid biosynthesis. In the fat body, main lipogenic organ in *Drosophila*, dSTING co-localized with both ACC and FASN in a cortical region of the ER. *dSTING* deletion resulted in the disturbed ACC localization in fat body cells and greatly reduced the activity of FASN in the in vitro assay.

Importantly, we also observed that ACC and FASN interacted with each other. Malonyl-CoA, the product of ACC, serves as a substrate for the FASN reaction of fatty acid synthesis. Enzymes that are involved in sequential reactions often physically interact with each other and form larger multi-enzyme complexes, which facilitates the substrate channeling and efficient regulation of the pathway flux (*Schmitt and An, 2017*; *Kastritis and Gavin, 2018*; *Sweetlove and Fernie, 2018*; *Zhang and Fernie, 2021*). There are several evidences of the existence of the multi-enzyme complex involved in fatty acid biosynthesis. ACC, ACL (ATP citrate lyase), and FASN physically associated in the microsomal fraction of rat liver (*Gillevet and Dakshinamurti, 1982*). Moreover, in the recent work, a lipogenic protein complex including ACC, FASN, and four more enzymes was isolated from the oleaginous fungus *Cunninghamella bainieri* (*Shuib et al., 2018*). It is possible that a similar multi-enzyme complex exists in *Drosophila* and other metazoan species, and it would be of great interest to identify its other potential members.

How does STING exerts its effect on lipid synthesis? Recently, the evidence has emerged for the control of the de novo fatty acid synthesis by two small effector proteins – MIG12 and Spot14. MIG12 overexpression in livers of mice increased total fatty acid synthesis and hepatic triglyceride content (*Kim et al., 2010*). It has been shown that MIG12 protein binds to ACC and facilitates its polymerization thus enhancing the activity of ACC (*Kim et al., 2010*; *Park et al., 2013*). For Spot14, both the activation and inhibition of de novo lipogenesis have been reported, depending upon the tissue type and the cellular context (*Rudolph et al., 2014*; *LaFave et al., 2006*; *Knobloch et al., 2013*). Importantly, there is an evidence that all four proteins – ACC, FASN, MIG12, and Spot14 – exist as a part of a multimeric complex (*McKean, 2016*). It is plausible to suggest that *Drosophila* STING plays a role similar to MIG12 and/or Spot14 in regulating fatty acid synthesis. We propose that dSTING might 'anchor' ACC and FASN possibly together with other enzymes at the ER membrane. The resulting complex facilitates fatty acid synthesis by allowing for a quicker transfer of malonyl-CoA product of ACC to the active site of FASN. In *dSTINGΔ* mutants, ACC loses its association with some regions of the ER resulting in the weakened interaction between ACC and FASN. We did observe less FASN

immunoprecipitated with ACC in *dSTINGΔ* mutants compared to control flies, and the opposite effect was found in flies expressing GFP-tagged dSTING (*Figure 4C*).

It has been shown that de novo synthesis of fatty acids continuously contributes to the total fat body TAG storage in *Drosophila* (*Heier and Kühnlein, 2018*; *Wicker-Thomas et al., 2015*; *Parvy et al., 2012*; *Garrido et al., 2015*). We hypothesize that the reduced fatty acid synthesis due to the lowered FASN enzyme activity in *dSTINGΔ* deletion mutants might be responsible for the decreased TAG lipid storage and starvation sensitivity phenotypes. Sensitivity to oxidative stress might also be explained by the reduced TAG level. Evidences exist that the lipid droplets (consisting mainly of TAGs) provide protection against reactive oxygen species (*Bailey et al., 2015*; *Jarc et al., 2018*; *Liu et al., 2015b*). Furthermore, flies with ACC RNAi are found to be sensitive to the oxidative stress (*Katewa et al., 2012*).

In addition to its direct role in ACC/FASN complex activity, STING might also affect a phosphorylation status of ACC and/or FASN. Both proteins are known to be regulated by phosphorylation/dephosphorylation (*Horton et al., 2002*; *Brownsey et al., 2006*; *Tong, 2005*; *Jin et al., 2010*). In mammals, STING is an adaptor protein that transmits an upstream signal by interacting with kinase TBK1 (TANK-binding kinase 1). When in a complex with STING, TBK1 activates and phosphorylates IRF3 allowing its nuclear translocation and transcriptional response (*Tanaka and Chen, 2012*; *Liu et al., 2015a*; *Zhong et al., 2008*). It is possible that in *Drosophila*, STING recruits a yet unidentified kinase that phosphorylates ACC and/or FASN thereby changing their enzymatic activity.

*Drosophila* STING itself could also be regulated by the lipid- synthesizing complex. STING palmitoylation was recently identified as a posttranslational modification necessary for STING signaling in mice (*Mukai et al., 2016*; *Hansen et al., 2019*; *Hansen et al., 2018*). In this way, palmitic acid synthesized by FASN might participate in the regulation of dSTING possibly providing a feedback loop.

The product of ACC – malonyl-CoA – is a key regulator of the energy metabolism (*Saggerson, 2008*). During lipogenic conditions, ACC is active and produces malonyl-CoA, which provides the carbon source for the synthesis of fatty acids by FASN. In *dSTING* knockout, FASN activity is decreased and malonyl-CoA is not utilized and builds up in the cells. Malonyl-CoA is also a potent inhibitor of carnitine palmitoyltransferase CPT1, the enzyme that controls the rate of fatty acid entry into the mitochondria, and hence is a key determinant of the rate of fatty acid oxidation (*McGarry and Brown, 1997*). Thus, a high level of malonyl-CoA results in a decreased fatty acid utilization for the energy. This might explain the down-regulation of lipid catabolism genes that we observed in *dSTINGΔ* mutants (*Figure 3C*). A reduced fatty acid oxidation in turn shifts cells to the increased reliance on glucose as a source of energy. Consistent with this notion, we observed an increased glucose level in fed *dSTINGΔ* mutant flies (*Figure 2D*), as well as increased levels of phosphoenolpyruvate (PEP) (*Figure 5—figure supplement 1*). PEP is produced during glycolysis, and its level was shown to correlate with the level of glucose (*Moreno-Felici et al., 2019*). A reliance on glucose for the energy also has a consequence of reduced incorporation of glucose into trehalose and glycogen for storage, and therefore, lower levels of these storage metabolites, which we observed (*Figure 2B,C*). To summarize, based on our findings, we propose a model presented in *Figure 8*, which suggests a direct involvement of dSTING in the regulation of lipid metabolism.

Recent studies show that in mammals, the STING pathway is involved in metabolic regulation under the obesity conditions. The expression level and activity of STING were upregulated in livers of mice with high-fat diet-induced obesity (*Bai et al., 2017*). STING expression was increased in livers from nonalcoholic fatty liver disease (NAFLD) patients compared to control group (*Luo et al., 2018*). In nonalcoholic steatohepatitis mouse livers, STING mRNA level was also elevated (*Xiong et al., 2019*). Importantly, STING deficiency ameliorated metabolic phenotypes and decreased lipid accumulation, inflammation, and apoptosis in fatty liver hepatocytes (*Iracheta-Vellve et al., 2016*; *Petrasek et al., 2013*; *Qiao et al., 2018*).

Despite the accumulating evidences, the exact mechanism of STING functions in metabolism is not completely understood. The prevailing hypothesis is that the obesity leads to a mitochondrial stress and a subsequent mtDNA release into the cytoplasm, which activates cGAS-STING pathway (*Bai et al., 2017*; *Bai and Liu, 2019*; *Yu et al., 2019*). The resulting chronic sterile inflammation is responsible for the development of NAFLD, insulin resistance, and type 2 diabetes. In this case, the effect of STING on metabolism is indirect and mediated by inflammation effectors. The data presented in the current study strongly suggest that in *Drosophila*, STING protein is directly involved

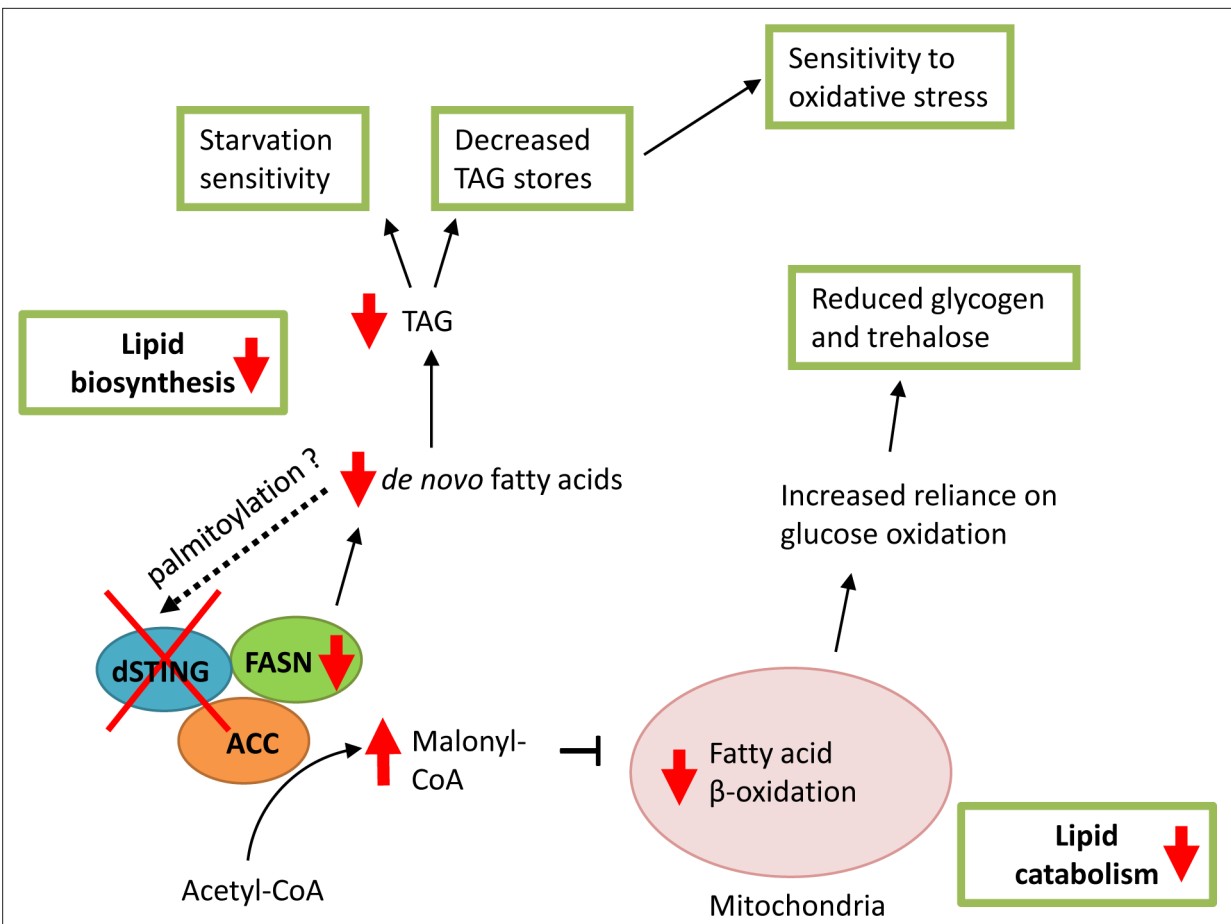

**Figure 8.** Model of *dSTING* deletion effect on *Drosophila* metabolism. Based on our data, dSTING interacts with lipid synthesizing enzymes acetyl-CoA carboxylase (ACC) and fatty acid synthase (FASN). In the absence of dSTING, the activity of FASN is reduced which results in decreased de novo fatty acid synthesis and triglyceride (TAG) synthesis. Low TAG level in turn lead to sensitivity to starvation and oxidative stress. Reduced FASN activity in *dSTING* mutants also results in ACC product malonyl-CoA build-up in the cells leading to the inhibition of the fatty acid oxidation in mitochondria. Reduced fatty acid oxidation shifts cells to the increased reliance on glucose as a source of energy resulting in reduced glycogen and trehalose levels in *dSTING* mutants. Palmitic acid synthesized by FASN might participate in the regulation of dSTING via palmitoylation possibly providing a feedback loop.

in lipid metabolism by interacting with the enzymes involved in a lipid biosynthesis. This raises the question if the observed interaction is unique for *Drosophila* or it is also the case for mammals. Future work is needed to elucidate the evolutionary aspect of STING role in metabolism. Understanding the relationships between STING and lipid metabolism may provide insights into the mechanisms of the obesity-induced metabolism dysregulation and thereby suggest novel therapeutic strategies for metabolic diseases.

## Materials and methods

**Key resources table**

| Reagent type (species) or resource | Designation | Source or reference | Identifiers | Additional information |
|---|---|---|---|---|
| Gene (*Drosophila melanogaster*) | *dSTING* | GenBank | FLYB: FBgn0033453 | |
| Genetic reagent (*D. melanogaster*) | *Sting*<sup>EY06491</sup> | Bloomington *Drosophila* Stock Center | BDSC: 16,729 RRID:BDSC_16729 | |

*Continued on next page*

*Continued*

| Reagent type (species) or resource | Designation | Source or reference | Identifiers | Additional information |
|---|---|---|---|---|
| Genetic reagent (*D. melanogaster*) | *yolk-Gal4* | Bloomington *Drosophila* Stock Center | BDSC: 58,814 RRID:BDSC_58814 | |
| Genetic reagent (*D. melanogaster*) | *cg-GAL4* | Bloomington *Drosophila* Stock Center | BDSC: 7,011 RRID:BDSC_7011 | |
| Genetic reagent (*D. melanogaster*) | *tub-GAL4* | Bloomington *Drosophila* Stock Center | BDSC: 5,138 RRID:BDSC_5138 | |
| Genetic reagent (*D. melanogaster*) | *dSTING-RNAi* | NIG-Fly, National Institute of Genetics, Japan | HMJ23183 | |
| Genetic reagent (*D. melanogaster*) | *dSTINGΔ; GFP-dSTING-WT* | This paper | | |
| Antibody | Anti-ACC (rabbit polyclonal) | Cell Signaling | Cat# 3,676 RRID:AB_2219397 | IF(1:200), WB(1:1000) |
| Antibody | anti-FASN (guinea pig polyclonal) | *Moraru et al., 2018* | | A gift from A.Teleman IF(1:150), WB(1:2000) |
| Antibody | anti-Calnexin (mouse monoclonal) | DHSB | Cat# Cnx99A 6-2-1 RRID:AB_2722011 | IF (1:30) |
| Antibody | anti-GFP (rabbit polyclonal) | Proteintech | Cat# 50430–2-AP RRID:AB_11042881 | IF (1:100) |
| Sequence-based reagent | CG1667-F | This paper | PCR primers | ATGGCAATCGCTAGCAACGT |
| Sequence-based reagent | CG1667-R | This paper | PCR primers | TGGCTACAATGCGAATAGAGGT |
| Commercial assay or kit | Acetyl-CoA Carboxylase assay kit | MyBioSource | Cat# MBS8303295 | |
| Chemical compound, drug | Nile Red | Thermo Fisher Scientific | Cat# N1142 | |
| Chemical compound, drug | HCS LipidTox Green | Thermo Fisher Scientific | Cat# H34475 | |
| Chemical compound, drug | Blue dye no. 1 | Millipore Sigma | Cat# 3844-45-9 | |
| Software, algorithm | GraphPad Prism | GraphPad Software | RRID:SCR_002798 | |

## *Drosophila* strains and genetics

Deletion mutations of *dSTING* gene (*dSTINGΔ*) were created by imprecise excision of *P* element-based transposon *P{EPgy2}Sting^EY06491*(FBti0039337). This transposon is mapped 353 bp upstream of the *dSTING* start codon. To initiate excision, males *y^1*,*w^67c23*; *P{w^+mC, y^+mDint2 = EPgy2} STING^EY0649*(Bloomington stock 16729) were crossed to females of the 'jump' stock *y^1w^1118*;*CyO, PBac{w^+mC=Delta 2–3. ExeI}2/amos^Tft*, bearing Δ2–3 transposase on a second chromosome, marked by *Curly*. F1 *Curly* males *y^1w^1118*; *P{w^+mC, y^+mDint2 = EPgy2} STING^EY0649/CyO, PBac{w^+mC = Delta 2–3. ExeI}* two were collected and crossed to *w^1118*; *If/CyO* females. The resulting F2 progeny was screened for white-eyed flies. White-eyed flies were crossed individually to *w^1118*; *If/CyO* to set up stocks *dSTINGΔ/CyO* and then *w^1118*; *dSTINGΔ/dSTINGΔ* homozygotes. The genomic DNA of these mutants was isolated. Mutations were confirmed by sequence determination following the PCR amplification with *dSTINGΔ* primer: 5'-CTCAGAATTCTCATTTATTCTGGCC-3'. RT-PCR analysis of *dSTING* expression confirmed that obtained deletions are *dSTING* null mutations.

For rescue experiments, *pCasper*-based vector containing UAS sequence followed by native *dSTING* promoter (437 bp upstream *dSTING* start codon) and GFP-tagged *dSTING* cDNA (clone #LP14056, DGRC, Bloomington) was injected into *w^1118* *Drosophila* embryos (Model System Injections, Raleigh, NC). Fly stocks *w^1118*; *dSTINGΔ/dSTINGΔ*; *GFP-dSTING-WT/GFP-dSTING-WT* were set up. The expression of tagged proteins was verified by immunoblot analysis with anti-GFP antibody.

For overexpression of GFP-dSTING in fat body for IP experiments, *cg-GAL4* driver was used (Bloomington stock 7011). For ubiquitous overexpression in mass-spec experiment, *tub-GAL4* driver was used (Bloomington stock 5138). For RNAi of *dSTING* in fat body, *yolk-GAL4* driver was used

(Bloomington stock 58814). Fly stock $y^1v^1;P\{TRiP.HMJ23183\}attP40/CyO$ (NIG-Fly, National Institute of Genetics, Japan) was used for *dSTING* RNAi experiments.

Flies were grown and maintained on food consisting of the following ingredients: one part of Nutri-Fly GF (Genesee Scientific, cat. 66–115) and three parts of Jazz-mix (Fisher Scientific, cat. AS153). All crosses were carried out at 25 °C. $w^{1118}$ fly stock was used as a wild-type control.

## Starvation and oxidative stress

For life span analysis, newly eclosed flies (females or males) were transferred to fresh food every 2 days, and dead flies were counted.

For starvation stress assay, 5 -day-old adult flies (females or males) were transferred from normal food to the vials containing Whatman filter paper soaked with PBS (15–20 flies per vial). Fresh PBS was added every 24 hr to prevent drying. Dead flies were counted every 12 hr.

For oxidative stress assay, 5 -day-old adult flies (females or males) were transferred from normal food to the vials containing normal food supplemented with 5 % hydrogen peroxide (15–20 flies per vial). Dead flies were counted every 12 hr.

For starvation stress resistance experiment on larvae, second-instar larvae (~53 hr) were transferred to the media containing 1.2% agarose. Surviving larvae were counted every 12 hr. For oxidative stress resistance, early third-instar larvae (~74 hr) were transferred to the media containing regular food supplemented with 10 mM paraquat. Percentages of pupae formed and imago eclosed were counted.

GFP-dSTING tissue expression *dSTINGΔ;GFP-dSTING* flies were used. From the third-instar larvae, the following tissues/organs were dissected: fat body, guts, brains (neural ganglia), and salivary glands. From the 5 -day-old adults, the following tissues were dissected: testes, ovaries, thoraxes, heads, guts, and abdominal carcasses. Abdominal carcass is what is left of the abdomen after the gut and testes/ovaries have been removed. Tissues/organs were placed in 1×  Laemmli buffer and boiled 5 min at 95 °C. 10 μg of extract was loaded per well of SDS–PAGE gel. Western blotting was performed using antibodies against GFP (1:1000, Santa Cruz Biotechnology, B2, cat. sc-9996).

## Axenic flies

To obtain axenic flies, 0–12 hr embryos were collected, dechorionated for 5 min in 50% Clorox, washed 2 × with autoclaved water, and transferred to sterile food. The axenity of flies was confirmed by PCR from flies homogenate using primers to 16 s rDNA gene (8FE, 5'- AGAGTTTGATCMTGGCTCAG-3' and 1492 R, 5'- GGMTACCTTGTTACGACTT-3').

## Triglycerides and glycogen quantifications

Eight 5 -day-old males (with heads removed) were collected, frozen in liquid nitrogen, and stored at –80 °C. Flies were ground in 200 μl of PBST buffer (PBS with 0.01 % Triton X-100) and heated at 70 °C for 10 min (*Tennessen et al., 2014*). For *yolk-GAL4* experiment, only females were used for TAG and glycogen measurement. In this case, six females (with heads removed) were used per sample.

For TAG measurement, 6 μl of homogenate were mixed with 25 μl of PBS and 30 μl of TAG reagent (Pointe Scientific, Cat. T7531) or Free Glycerol Reagent (MilliporeSigma, Cat. F6428). Triglyceride standard solution (from Pointe Scientific, Cat. T7531 kit) and glycerol standard solution (MilliporeSigma, Cat. G7793) were used as standards. Reactions were incubated for 30 min at 37 °C, centrifuged 6000 g for 2 min, and supernatants were transferred to 96-well plate, after which absorbance was read at 540 nm. The TAG concentration in each sample was determined by subtracting the values of free glycerol in the corresponding sample. Total protein level in the samples was determined using Bio-Rad Protein Assay Dye Reagent Concentrate (Bio-Rad, Cat. 5000006).

For glycogen measurement, homogenate was centrifuged 5 min at 10,000 g. 6 μl of supernatant were mixed with 24 μl of PBS and 100 μl of glucose reagent (MilliporeSigma, Cat. GAGO20) with or without the addition of amyloglucosidase (MilliporeSigma, Cat. A1602, 0.25U per reaction) and transferred to 96-well plate. Glycogen solution (Fisher Scientific, Cat. BP676-5) and glucose solution (MilliporeSigma, Cat. 49161) were used as standards. Reactions were incubated 60 min at 37 °C, after which 100 μl of sulfuric acid were added to stop the reaction, and the absorbance was read at 540 nm. Glycogen concentration in each sample was determined by subtracting the values of free glucose in corresponding sample. Total protein level in the samples was determined using Bio-Rad Protein Assay Dye Reagent Concentrate (Bio-Rad, Cat. 5000006).

## Hemolymph sugar quantification

Fifty 5 -day-old males were anesthetized with $CO_2$ and pricked with a needle in the thorax. 0.2 ml PCR tubes with caps removed were inserted inside 1.5 ml tube. Pricked flies were placed into a spin column (Zymo Research, Cat. N. C1005-50) with plastic ring and filling removed (leaving only bottom glass wool layer). Spin columns were inserted into a 1.5 ml tube with PCR tube, centrifuged 5 min at 2500 g at 4°C, shaken to dislodge flies, and centrifuged one more minute. 0.5 μl of collected hemolymph were mixed with 4.5 μl of PBS, heated at 70 °C for 5 min, centrifuged at 6000 g 15 s, and placed on ice. To measure glucose level, 2 μl samples (in duplicates) were mixed with 100 μl Infinity glucose reagent (Thermo Scientific, Cat. N. TR15421) in a 96-well plate, and after 5 min incubation at 37 °C, the absorbance was read at 340 nm. To measure trehalose level, 1 μl of trehalase (MilliporeSigma, Cat. No T8778) was added to the wells with measured glucose (see above). Plate was incubated at 37 °C overnight, the absorbance was read at 340 nm, and glucose readings were subtracted from obtained values. Total protein level in the samples was determined using Bio-Rad Protein Assay Dye Reagent Concentrate (#5000006, Bio-Rad).

## CAFE assay

CAFE assay was adopted from *Diegelmann et al., 2017*. Plastic bottles with carton caps and small holes on the bottom to allow for air circulation were used. Five openings were made in a carton cap to fit the pipette tips of 2–20 μl volume. Five glass capillaries (Drummond Scientific Company, Cat. No. 2-000-001) were filled with 5 μl of 20 % sucrose solution in water and inserted into pipet tips on the cap. Ten 4 -day-old males were placed in each bottle, and all bottles were placed into a plastic box containing wet paper towel to provide humidity. Control bottles that contained no flies were set up to account for liquid evaporation. After 24 hr and 48 hr, the amount of food consumed in each bottle was measured as follows: Food consumption (μl/fly) = (Food uptake (μl) − Evaporative loss (μl))/total number of flies in the vial.

## Smurf gut permeability assay

5 -day-old flies were transferred from normal food to food containing 2.5 % (wt/vol) Blue dye no. 1 (MilliporeSigma, Cat. No 3844-45-9). Flies were kept on dyed food for 12 hr. A fly was counted as a Smurf if dye coloration could be observed outside of the digestive tract.

## Lipid droplet staining

For Nile Red staining, adult fat bodies and guts were dissected in PBS, fixed in 4% paraformaldehyde for 20 min, washed twice with PBS, and mounted in fresh Nile Red solution with DAPI (0.5 mg/ml Nile Red [ThermoFisher, cat N1142] stock solution diluted 1000× with PBS supplemented with 30 % glycerol). For LipidTox staining, adult fat bodies were dissected in PBS, fixed in 4% paraformaldehyde for 20 min, washed once with PBST and twice with PBS, and stained with 50 × dilution of HCS LipidTox-Green (ThermoFisher, cat. H34475) in PBS. After LipidTox staining, fat bodies were washed with PBS, stained with DAPI, and mounted in Fluoromount-G (SouthernBiotech, cat. 0100–01). Images were collected using Olympus Fluoview FV3000. Quantification of surface area occupied by lipid droplets was performed using cellSens Dimension Desktop (Olympus). Minimum 8 (guts) or 11 (fat bodies) samples per genotype were analyzed.

## Microarray analysis

Total RNA was extracted from ten 5-day-old male flies (*w^1118* or *dSTINGΔ*), fed or 24 hr starved, using ZR Tissue and Insect RNA MicroPrep (Zymo Research, #R2030) according to the manufacturer's instructions. Three replicates per genotype/condition were used. Microarray analysis was performed at the Boston University Microarray and Sequencing Resource Core Facility. *Drosophila* Gene 1.0 ST CEL files were normalized to produce gene-level expression values using the implementation of the Robust Multiarray Average (RMA) (*Irizarry et al., 2003*) in the *affy* package (version 1.48.0) (*Gautier et al., 2004*) included in the Bioconductor software suite (version 3.2) (*Gentleman et al., 2004*) and an Entrez Gene-specific probeset mapping (20.0.0) from the Molecular and Behavioral Neuroscience Institute (Brainarray) at the University of Michigan (*Dai et al., 2005*). Array quality was assessed by computing Relative Log Expression (RLE) and Normalized Unscaled Standard Error (NUSE) using the *affyPLM* package (version 1.46.0). PCA was performed using the *prcomp* R function with expression

values that had been normalized across all samples to a mean of zero and a standard deviation of one. Differential expression was assessed using the moderated (empirical Bayesian) *t* test implemented in the *limma* package (version 3.26.9) (i.e., creating simple linear models with *lmFit*, followed by empirical Bayesian adjustment with *eBayes*). Correction for multiple hypothesis testing was accomplished using the Benjamini–Hochberg false discovery rate (FDR) (*Benjamini et al., 2001*). Human homologs of fly genes were identified using HomoloGene (version 68). All microarray analyses were performed using the R environment for statistical computing (version 3.2.0).

Gene Ontology (GO) analysis was conducted using the DAVID Functional Annotation Tool (https://david.ncifcrf.gov/).

GSEA (version 2.2.1) (*Subramanian et al., 2005*) was used to identify biological terms, pathways, and processes that are coordinately up- or down-regulated within each pairwise comparison. The Entrez Gene identifiers of the human homologs of the genes interrogated by the array were ranked according to the *t* statistics computed for each effect in the two-factor model and for each pairwise comparison. Any fly genes with multiple human homologs (or vice versa) were removed prior to ranking, so that the ranked list represents only those human genes that match exactly one fly gene. Each ranked list was then used to perform pre-ranked GSEA analyses (default parameters with random seed 1234) using the Entrez Gene versions of the Hallmark, Biocarta, KEGG, Reactome, Gene Ontology (GO), and transcription factor and microRNA motif gene sets obtained from the Molecular Signatures Database (MSigDB), version 6.0 (*Subramanian et al., 2007*).

## RT-qPCR

RNA was isolated from eight 5 -day-old males using ZR Tissue and Insect RNA MicroPrep (Zymo Research, #R2030). DNA was removed using TURBO DNAse (Invitrogen, #AM2238) following manufacturer's recommendations. cDNA was generated from 1 µg of total RNA using ProtoScript II First Strand cDNA Synthesis Kit (New England Biolabs, E6560). RT-qPCR analysis was performed in Luna Universal qPCR Master Mix (New England Biolabs, #M3003) using a Roche LightCycler480 (Roche). Primers used were as follows: CG1667-F, 5'-ATGGCAATCGCTAGCAACGT-3' and CG1667-R, TGGCTACAATGCGAATAGAGGT (*Hu et al., 2013*). Two qPCR technical replicates were conducted for three-four biological replicates. Relative expression was normalized to *rpl32* reference gene using ΔΔCt comparative method.

## Mass spectrometry

Fat body from six third-instar larvae ubiquitously overexpressing GFP-dSTING (genotype *w^1118^;+/+;tub-GAL4/GFP-dSTING*) or control larvae (genotype *w^1118^*) were ground in 200 µl of IP buffer (25 mM HEPES, pH 7.6, 0.1 mM EDTA, 12 mM $MgCl_2$, 100 mM NaCl, 1% NP-40) and extracted for 30 min at RT. Recombinant GFP protein was added to the control lysate. Samples were centrifuged 10,000 g for 5 min, and supernatant was precleared with 20 µl of protein G sepharose beads (Amersham Biosciences, cat. 17-0618-01) for 2 hr at 4 °C. Precleared lysate was incubated with 4 µg of antibodies against GFP tag (DSHB, 4C9) overnight at 4 °C. Beads were washed four times with IP buffer, and immunoprecipitation reactions were separated by SDS–PAGE and most prominent individual gel bands corresponding to ~250 kDa and ~30 kDa were excised. Mass spectromery detection was performed at The Proteomics Resource Center at The Rockefeller University. Proteins were reduced with DTT, alkylated with iodoacetamide, and trypsinized. Extracted peptides were analyzed by nanoLC-MS/MS (Dionex 3,000 coupled to Q-Exactive+, Thermo Scientific), separated by reversed phase using an analytical gradient increasing from 1 % B/ 99 % A to 40 % B/ 60 % A in 27 min (A: 0.1 % formic acid, B: 80 % acetonitrile/0.1 % formic acid). Identified peptides were filtered using 1 % FDR and Percolator (*Käll et al., 2007*). Proteins were sorted out according to estimated abundance. The area is calculated based on the most abundant peptides for the respective protein (*Silva et al., 2006*). Proteins not detected or present in low amounts are assigned an area zero. Data were extracted and queried against Uniprot *Drosophila* using Proteome Discoverer and Mascot.

## Immunoprecipitation

Fifteen abdomens of 5 -day-old males were ground in 300 µl of IP buffer (10 mM Tris pH 7.4, 1 mM EDTA, 1 mm EGTA, 2 mM $MgCl_2$, 2 mM $MnCl_2$, 1× Halt Protease, and Phosphatase Inhibitor Cocktail [Thermo Scientific, cat. 78446], supplemented with 100 mM NaCl, 0.02 % Triton X-100 for ACC IP

and 150 mM NaCl, 0.1 % Triton X-100 for FASN IP). After extraction for 30 min at RT, samples were centrifuged 600 g at 3 min and supernatants were precleared with 15 µl of protein A agarose beads (Goldbio, cat. P-400–5) for 2 hr at RT. After discarding the beads, supernatant was divided in half and incubated with either antibodies or corresponding normal IgG overnight at 4 °C. Antibodies used were as follows: rabbit anti-ACC (Cell Signaling, #3676), guinea pig anti-FASN (generously provided by A.Teleman [*Moraru et al., 2018*]), rabbit IgG (Sino Biological, cat. CR1), and guinea pig IgG (Sino Biological, cat. CR4). Beads were washed three times with IP buffer, and bound proteins were analyzed by SDS–PAGE and western blotting.

## ACC activity assay

Assay was conducted using acetyl-CoA carboxylase assay kit (#MBS8303295, MyBioSource). Eight males were collected, frozen in liquid nitrogen, and stored at –80 °C. Flies were ground in 250 µl of assay buffer after which another 250 µl of assay buffer were added (total lysate volume 500 µl). Lysates were centrifuged 8,000 g for 10 min at 4 °C, and 300 µl of supernatant were transferred to a new tube. To set up the reaction, 10 µl of supernatant (or assay buffer for control reactions) were mixed with 90 µl of substrate and incubated 30 min at 37 °C after which the reactions were centrifuged 10,000 g for min at 4 °C. 5 µl of supernatant, water (for blank reaction), or standards (phosphate) were added to 100 µl of dye working reagent in a 96-well plate, and the absorbance at 635 nm was recorder after 5 min of incubation. Total protein level in the samples was determined using Bio-Rad Protein Assay Dye Reagent Concentrate (#5000006, Bio-Rad). One unit of ACC activity is defined as the enzyme generates 1 nmol of $PO_4^{3-}$ per hour.

## FASN activity assay

Assay was conducted essentially as described in *Moraru et al., 2018*. Eight males were collected, frozen in liquid nitrogen, and stored at –80 °C no more than 1 day. Flies were ground in 150 µl of homogenization buffer (10 mM potassium phosphate buffer pH 7.4, 1 mM EDTA, 1 mM DTT) and 300 µl of cold saturated ammonium sulfate solution (4.1 M in water, pH 7) were added to the lysate. After incubation on ice for 20 min, samples were centrifuged at 20,000 g for 10 min at 4 °C and supernatant was carefully removed. Pellet was resuspended in 200 µl of homogenization buffer, centrifuged 10,000 g 10 min, and 150 µl of supernatant were transferred to a new tube. To set up the reaction, 20 µl of sample were added to 160 µl of 0.2 mM NADPH (#9000743, Cayman Chemical) in 25 mM Tris pH 8.0 and incubated 10 min at 25 °C in a 96-well plate. 20 µl of water (for control reaction) or a mix of 10 µl of 0.66 mM acetyl CoA (#16160, Cayman Chemical) and 10 µl of 2 mM malonyl-CoA (#16455, Cayman Chemical) were added to the reaction, and absorbance at 340 nm was recorded every 5 min for 60 min at 25 °C using Synergy two multi-mode microplate reader (BioTec). Absorbance for control reaction was subtracted for each time point. Total protein level in the samples was determined using Bio-Rad Protein Assay Dye Reagent Concentrate (#5000006, Bio-Rad).

## Polar metabolite profiling

For polar metabolite profiling experiment, twenty 5 -day-old adult flies (males) were collected in Eppendorf tube, weighted, frozen in liquid nitrogen, and stored at –80 °C. For metabolite extraction, flies were transferred to 2 ml tubes with 1.4 mm ceramic beads (Fisher Scientific, cat. 15-340-153), 800 µl of extraction buffer was added (80 % methanol [A454, Fisher Scientific] and 20% $H_2O$ [W7SK, Fisher Scientific], standards), and flies were processed on BeadBlaster 24 Microtube Homogenizer (Benchmark Scientific) at 6 m/s for 30 s. Tubes were incubated on rotator for 1 hr at 4 °C and centrifuged 20,000 g 15 min at 4 °C. Seven hundred microliters of supernatant was transferred to new Ependorf tube and dried in a vacuum centrifuge.

Metabolomics analysis was performed at The Proteomics Resource Center at The Rockefeller University. Polar metabolites were separated on a ZIC-pHILIC 150 × 2.1 mm (5 µm particle size) column (EMD Millipore) connected to a Thermo Vanquish ultrahigh-pressure liquid chromatography (UPLC) system and a Q Exactive benchtop orbitrap mass spectrometer equipped with a heated electrospray ionization (HESI) probe. Dried polar samples were resuspended in 60 µl of 50 % acetonitrile, vortexed for 10 s, and centrifuged for 15 min at 20,000 g at 4 °C, and 5 µl of the supernatant was injected onto the LC/MS system in a randomized sequence. Mobile phase A consisted of 20 mM ammonium carbonate with 0.1 % (vol/vol) ammonium hydroxide (adjusted to pH 9.3), and mobile phase B was

acetonitrile. Chromatographic separation was achieved using the following gradient (flow rate set at 0.15 ml min⁻¹): gradient from 90% to 40% B (0–22 min), held at 40 % B (22–24 min), returned to 90 % B (24–24.1 min), equilibrating at 90 % B (24.1–30 min). The mass spectrometer was operated in polarity switching mode for both full MS and data-driven aquisition scans. The full MS scan was acquired with 70,000 resolution, $1 \times 10^6$ automatic gain control (AGC) target, 80 ms max injection time, and a scan ranges of 110–755 m/z (neg), 805–855 m/z (neg), and 155–860 m/z (pos). The data-dependent MS/MS scans were acquired at a resolution of 17,500, $1 \times 10^5$ AGC target, 50 ms max injection time, 1.6 Da isolation width, stepwise normalized collision energy of 20, 30, and 40 units, 8 s dynamic exclusion, and a loop count of 2, scan range of 110–860 m/z.

Relative quantitation of polar metabolites was performed using Skyline Daily56 (v.20.1.1.158) with the maximum mass and retention time tolerance were set to 2 ppm and 12 s, respectively, referencing an in-house library of chemical standards. Metabolite levels were normalized to the total protein amount for each condition.

## Membrane and cytoplasmic protein extraction

Membrane fractionation was performed following the protocol from *Abas and Luschnig, 2010* with modifications. Flies were either fed or starved for 24 hr. Thirteen abdomens (guts and testes removed) from 5 -day-old males of flies expressing GFP-dSTING (*w¹¹¹⁸;dSTINGΔ/dSTINGΔ;GFP-dSTING/GFP-dSTING*) were ground in 100 µl of EB (30 mM Tris pH 7.5, 25 % sucrose, 5 % glycerol, 5 mM EDTA, 5 mM EGTA, 5 mM KCl, 1 mM DTT, aprotinin, leupeptin, PMSF), spun down at 600 g for 3 min to remove debris. Supernatant after centrifugation represents total protein fraction. Supernatant was diluted twice with 100 µl $H_2O$ and centrifuged at 21,000 g for 2 hr at 4 °C. Resulting supernatant represents cytoplasmic fraction. Pellet was resuspended in 30 µl of EB supplemented with 0.5 % Triton X-100, resulting in membrane fraction sample. Proteins were subjected to SDS–PAGE and western blotting. Total protein fraction was used for assessing the levels of ACC and FASN. Cytoplasmic and membrane fractions were used to analyze GFP-dSTING localization. Antibodies used were as follows: ACC (1:1000, C83B10, Cell Signaling, #3676), FASN (1:2000, generously provided by A.Teleman [*Moraru et al., 2018*]), Gapdh1 (1:2000, Sigma-Aldrich, #G9545), ATPβ (1:1000, Abcam, cat. ab14730), and GFP (1:1000, Santa Cruz Biotechnology, B2, cat. sc-9996).

## Immunostaining

Adult fat body and guts were dissected in PBS, fixed in 4 % paraformaldehyde for 20 min, washed with PBST (PBS supplemented with 0.1 % Triton X-100), and blocked with PBST supplemented with 10 % goat serum for 1 hr at RT. Tissue were stained with primary antibodies in PBST +10 % goat serum overnight at 4 °C, washed three times with PBST, and incubated with secondary antibodies in PBST + 10 % goat serum for 2 hr at RT. Antibodies used were as follows: ACC (1:200, C83B10, Cell Signaling, #3676), Calnexin (1:30, DSHB, Cnx99A, 6-2-1-s), FASN (1:150, generously provided by A.Teleman [*Moraru et al., 2018*]), and GFP (Proteintech, cat. 50430–2-AP). After three washes with PBST, tissues were stained with DAPI, washed with PBS, and mounted in Fluoromount-G (SouthernBiotech, cat. 0100–01). Images were collected using Olympus Fluoview FV3000.

## Data analysis

All data are reported as mean ± SD. To determine statistical differences, Student's t-test was performed for comparison of two groups, and two-way ANOVA followed by Tukey multiple comparison test was utilized when three and more groups were compared. A probability value of $p<0.05$ was considered significantly different. Statistical calculations were performed using the GraphPad Prism software (La Jolla, CA). Survival curves were plotted and analyzed by log-rank analysis (Kaplan–Meier method) using GraphPad Prism software (La Jolla, CA).

## Acknowledgements

We would like to thank Adam Gower from the Boston University Microarray and Sequencing Resource Core Facility. All research materials and data from our studies will be freely available to other investigators. This work was supported by a grant from NIH to IC (GM121449).

## Additional information

### Funding

| Funder | Grant reference number | Author |
|---|---|---|
| National Institute of General Medical Sciences | GM121449 | Katarina Akhmetova Maxim Balasov Igor Chesnokov |

The funders had no role in study design, data collection and interpretation, or the decision to submit the work for publication.

### Author contributions

Katarina Akhmetova, Formal analysis, Investigation, Methodology, Resources, Validation, Visualization, Writing - original draft, Writing - review and editing; Maxim Balasov, Formal analysis, Methodology, Writing - review and editing; Igor Chesnokov, Conceptualization, Data curation, Funding acquisition, Methodology, Project administration, Supervision, Writing - review and editing

### Author ORCIDs

Katarina Akhmetova (ID) http://orcid.org/0000-0003-2475-3288
Igor Chesnokov (ID) http://orcid.org/0000-0002-6659-2913

### Decision letter and Author response

Decision letter https://doi.org/10.7554/eLife.67358.sa1
Author response https://doi.org/10.7554/eLife.67358.sa2

## Additional files

### Supplementary files

• Supplementary file 1. Microarray analysis of *Drosophila STING* mutant flies.
• Supplementary file 2. Mass spectrometry analysis of GFP-dSTING-interacting proteins.
• Transparent reporting form

### Data availability

Microarray data have been deposited in GEO under accession code GSE167164.

The following dataset was generated:

| Author(s) | Year | Dataset title | Dataset URL | Database and Identifier |
|---|---|---|---|---|
| Akhmetova KA, Gower AC | 2021 | Microarray data have been deposited in GEO under accession code GSE167164 | https://www.ncbi.nlm.nih.gov/geo/query/acc.cgi?acc=GSE167164 | NCBI Gene Expression Omnibus, GSE167164 |

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
