## [Decision Letter]

**Acceptance summary:**

This paper presents an interesting new facet of the function of STING in relation to its ability to regulate metabolism in animals.

**Decision letter after peer review:**

Thank you for submitting your article "*Drosophila* STING protein has a role in lipid metabolism" for consideration by *eLife*. Your article has been reviewed by 3 peer reviewers, one of whom is a member of our Board of Reviewing Editors, and the evaluation has been overseen by K VijayRaghavan as the Senior Editor. The reviewers have opted to remain anonymous.

Essential revisions:

1. A central message of this study is a direct role of dSTING in regulating metabolism, independent of its role in immunity. The potential chronic infection status of the flies in the experiments presented contributing to the metabolic changes observed is an important concern – reduced immunity in dSTING mutants could produce a higher pathogen load, which would consume energy. Please perform the storage metabolite assays (TAG, glycogen) under axenic conditions would help show that the two are or are not linked.

2. Cg-Gal4 expresses in fat body/circulation as well as lymph gland hemocytes. yolk-gal4 (adult-specific fat body driver) should be used to exclude the contribution of hemocytes. This is mandatory to exclude the contribution of non-fat body cells towards the phenotype.

3. In the manuscript experiments on both adult and larval fat body are shown in various figures. Reasons for the stages used and timeline of the experiments need to be stated clearly. The larval and adult stages cannot be used interchangeably. Assuming that the larval and adult fat body are equivalent is not advisable. Uniformity of the experimental set up is mandatory.

4. Spatiotemporal analysis of STRING across tissues is needed to further endorse/support the function proposed.

5. Among the changes in various metabolites presented, only that in TAG is convincing; others such as trehalose are less compelling. When comparing fed v starved in any genotype or between two genotypes, it would be best to show the changes in levels of a metabolite as a fractional/proportional change and then compare changes in these between genotypes.

6. Technical issues that must be addressed

(i) All survival curves are discontinuous data and authors should not present them as continuous data as in all the figures of the current version of the manuscript.

(ii) genetic and biochemical (IP, mass spec) controls need to be explicit for all experiments, including genetic background controls, and measuring the degree of expression when overexpressing and after knockdown.

(iii) A genetic rescue is missing in many cases, needs to be done to add robustness to the hypothesis. Genetic rescue of at least one larval phenotype with the GFP-STING is required.

(iv) The expression levels of dSTING-GFP in genotypes where it is used to rescue mutant phenotypes must be presented.

(v) The effectiveness of the RNAi line used in this study needs to be shown experimentally by RT-PCR.

*Reviewer #1:*

This paper presents the generation of a mutant allele of STING and a study of its impact on phenotypes linked to metabolism in *Drosophila* including sensitivity to starvation and oxidative stress. The authors find that altered sensitivity to starvation and oxidative stress is accompanied by changes in the levels of neutral fat (triglyceride) storage in the fat body, a key metabolic organ in *Drosophila*, thought to be equivalent to the lever and adipose tissue in mammals. To understand the basis of this phenotype, the authors performed a gene expression analysis and report alterations in the expression of immune response genes (as previously reported) and metabolic processes such as fatty acid metabolism. Metabolomics analysis indicates alterations in a range of cellular metabolites in STING mutants. Finally, they find that two enzymes that are essential for fatty acid synthesis engage in protein-protein interactions and co-localize with STING in fat body cells. Depletion of STING results in the mis-localization of one of the enzymes ACC and reduced activity of the other, FAS.

If STING is a direct regulator of metabolism, it would be interesting to know how it couples cellular state and environmental cues to fatty acid biosynthesis. These remain to be discovered and a broader role for STING in regulating fatty acid metabolism outside the fat body also awaits discovery and will be stimulated by this study.

Comments for the authors:

1. What is the expression pattern of STING across tissues and developmental times in *Drosophila*? Does this support the function proposed here ?

2. All survival curves are discontinuous data and authors should not present them as continuous data as in all the figures of the current version of the manuscript.

3. Among the changes in various metabolites presented, only that in TAG is convincing; others such as trehalose are less compelling. When comparing fed v starved in any genotype or between two genotypes, it would be best to show the changes in levels of a metabolite as a fractional/proportional change and then compare changes in these between genotypes.

4. "Together, our results demonstrate a direct role of STING in lipid metabolism in *Drosophila*."

This statement in the abstract is too strong given the data available in the manuscript.

5. Trehalose is synthesized in the fat body from glucose. Given the noted reduction in trehalose levels and elevation in glucose levels, is trehalose synthesis affected in STING mutants ?

6. GFP-dSTING construct: Please show the level of expression of this construct when expressed in mutant background relative to wild type sting levels.

7. Overexpression of STING had no effect given its proposed role as a direct regulator of metabolism. Related to the point above.

8. (Pospisilik et al., 2010) (Baumbach et al., 2014) "whereas in our study dSTING was absent from the very beginning of the development. "

This indicates a developmental role for STING which is unclear.

9. "Together, these data indicate that a deletion of dSTING results in a disruption of lipid/carbohydrate energy balance." This statement does not appear substantiated.

10. An important point to note is that the GAL4 driver used here expresses in fat body, hemocytes and lymph gland. This needs to be addressed since we are talking about a molecule that already has a demonstrated role in immune function. Otherwise, it will be difficult to uncouple the immune and metabolic role of STING.

11. The effectiveness of the RNAi line used in this study needs to be shown experimentally by RT-PCR.

12. The role of dSTING independent of the INR/Foxo pathway, as presented here is not at all convincing. See also details on specific figures below.

13. The IP and mass spec on dSTING. What are the controls and what did you do to ensure that the interactions seen are likely to be specific?

14. Many times, the paper switches between adults and larvae for expts and also uses adults and larvae at various points interchangeably. For e.g mas spec done from larval FB and confirmatory IP from adult abdomen. This is not advisable as the phenotypes presented here likely have a developmental origin. IT cannot also be assumed that the larval and adult fat body are equivalent.

15. Supp Figure 7. Many large changes in metabolomics data left unexplained and not discussed.

16. Given the already well-known role of STING in mammals the authors should present data on:

(i) Any changes in mitochondria in dSTING mutants?

(ii) Any evidence for DNA damage in dSTING mutants?

(iii) Strengthen the data that the metabolic changes seen in dSTING are not an indirect outcome of ongoing infection. Current one data is not strong enough.

17. Figure 2E' and 2F'. Quantification not clear. What do you mean by area of lipid droplets? Not related to conclusion claimed.

18. Suppl Figure 1C – quantification needed.

Suppl Figure 3 A. not clear what is 24 hrs and 48 hrs.

Suppl Figure 3B not clear.

19. Suppl Figure 6. The fractionation of cells and the purity of the fractions not shown. Controls needed here. Fraction markers as in Figure S8 needed. Data presented here is not convincing. Positive control also needed.

*Reviewer #2:*

Akehmetova et al. intend to demonstrate an unappreciated role of dSTING in lipid metabolism. Although dSTING deleted flies show no error in metabolism, they are sensitive to starvation and oxidative stress. The authors discovered that upon starvation/ oxidative stress, there is a significant decrease in the storage metabolites, such as TAG, trehalose and glycogen in dSTING deleted flies.

They identified two essential fatty-acid biosynthesis enzymes – Acetyl-CoA carboxylase (ACC) and fatty acid synthase (FAS) – as interacting partners through molecular and biochemical approaches for dSTING.

They propose that dSTING, FAS and ACC interacted with each other, and probably all three proteins are components of a large complex. In the absence of dSTING, there is a decrease in FAS activity and defects in ACC cellular localization, suggesting a direct role of dSTING in the lipid metabolism of fruit flies.

These data add an interesting insight into our understanding of stress-induced altered metabolism. Data mostly well support this paper's conclusions, but some aspects of image acquisition and data analysis need to be clarified and extended. Following points should be addressed.

1) Title: "*Drosophila* STING protein has a role in lipid metabolism". The title is misleading as a total loss of STING does not have a phenotype. The phenotypes are detectable upon stress. It emphasizes the fact that the dSTING function enables the organism to tide over stress. This thought should be reflected in the title.

2) The timeline for starvation/ oxidative stress should be schematically present or mentioned at least once in the text.

3) the dSTING starved larvae do they have a delay in pupation? Do the pupae eclose? Since pupae is a non-feeding state, the effect that we see in larvae will be nice to extend in the pupal stage.

4) "The expression of GFP-dSTING partially or entirely rescued the sensitivity of dSTINGΔ deletion flies to both starvation and oxidative stress (Figure 1B,C), suggesting that observed phenotypes are caused by dSTING deficiency. " Can it rescue larval defects…it needs to be shown.

5) "the age-related mortality was slightly reduced, especially for females (Figure 1D)". A possible explanation needs to be mentioned.

6) "Two RNAi screens for obesity and anti-obesity genes in *Drosophila* did not reveal any significant changes in TAG level in dSTING-deficient flies (Pospisilik et al., 2010) (Baumbach et al., 2014). The potential discrepancy with our data might be explained by the fact that in both mentioned studies RNAi was induced only 2-8 days after eclosion, whereas in our study dSTING was absent from the very beginning of the development" – Question: was the starvation in the current study not limited to the adulthood? To avoid confusion, the regime of starvation, a scheme is needed. If RNAi is done from the very beginning, will it endorse the current findings?

That might resolve the discrepancy.

7) Additional marker for neutral lipid should be used (LipidTOX).

8) "fat body specific RNAi of dSTING resulted in starvation sensitivity and reduced TAG level in adult flies (Supplementary Figure 4), highlighting the role of dSTING in fat body functions." Cg-Gal4 expresses in also in blood/hemocytes. A more specific driver for the fat body like yolk-gal4 (adult-specific) should be used to exclude the contribution of hemocytes.

9) "Consequently, we did not observe the increase in nuclear FoxO which is a characteristic of FoxO activation (Supplementary Figure 6)". Upon starvation, both dSTING lof and control upregulated nuclear FoxO

compared to fed.…any comments?

10) (Figure 5C, Supplementary Figure 7) Please mention the loading control, not as pnut but peanut or cpr=NADPH-cytochrome P450 reductase in the figure legend or the panel itself.

11) "Both ACC and FAS partially co-localized with GFP-dSTING at the cell periphery region of the ER (Figure 6B,C)". The Calnexin labeling performed is not convincing.

It needs to be repeated and a higher magnification to be included.

12) 7A: higher magnification to be included.

13) Strangely enough, how can ACC activity remain unchanged even though it is mislocalized upon loss of dSTING? ACC localization is not essentially related to its activity?

14) The FAS protein is normal, but activity is decreased. Is it due to its partial colocalization with the aggregates of ACC in the absence of STRING? It needs to be discussed.

15) "Reliance on glucose for energy also has a consequence of reduced incorporation of glucose into trehalose and glycogen for storage, and therefore, lower levels of these storage metabolites, which we observed (Figure 2B,C)" – what happens if we inhibit glucose by 2DG or genetically in String loss: will there be an increase in the levels of glucose into trehalose and glycogen?

*Reviewer #3:*

Akhmetova et al. describe a novel role for an immune protein, dSTING, in the regulation of lipid metabolism. The *Drosophila* STING protein is required for energy storage in triglycerides, glycogen, and trehalose. The mechanism underlying dSTING's effects on lipogenesis appear to include direct interactions between dSTING and the lipogenic enzymes fatty acid synthase and acetyl coA carboxylase. Loss of dSTING had surprising effects on gene expression, including that of FASNs and ACC. How dSTING affects carbohydrate metabolism is unclear, but may be an indirect result of reducing the conversion of acetyl carbons into lipids. The direct effect of an immune protein on lipid homeostasis is a novel finding. One strength is that the authors use a broad range of approaches (expression profiling, microscopy, protein and metabolite mass spectrometry, co-immunoprecipitation, and enzymology) to develop a fascinating and impressive model of how dSTING controls lipogenesis.

I really liked this story – it's one of the most exciting recent findings in the field and it links immunity and metabolism in an unexpected way. There is a stunning amount of work that explores many facets of dSTING function and taken together, I found their data quite compelling. The work will be of interest to those studying immunity, immunometabolism, and metabolism. It serves as an example of how protein localization can affect metabolism and is also a potential platform to study how lipid metabolism can affect gene expression and carbohydrate flux.

There are some flaws in the experimental design and analyses that are of concern and prevent me from recommending acceptance.

1. The genetic background of the control flies should match the mutant alleles being used. For the deletion, there should be a precise excision of the P element used to generate the deletion, so that the genetic background is matched between the two. The genetic background of this insertion is not *w^1118^* but y[1] w[67c23]. For UAS crosses, the correct control should be a cross between the driver and an empty insertion site and/or the UAS crossed into the driver background. It is not clear to me if this was used consistently. The *w^1118^* stock control used throughout the paper is not the correct control and therefore it is difficult to interpret your results.

2. For some experiments, there seem to be low replicate #s (3-5) and the bar graphs showing the means instead of individual data points make it difficult to determine whether the data are normally distributed. Scatter plots with each data point would help the reviewers better assess the rigor of these experiments.

3. The survival analyses in 1B, C, D and elsewhere should use Mantel-Cox statistics, not t tests. And comparisons of more than two sample types should probably use an ANOVA, not a student's t-test.

4. It would also be great if quantitative data were presented for the Western blots. A couple of the blots seem unevenly loaded (5B', S6 fed) and in Figure S6, a nuclear protein should have been used as a loading control for nuclear fractions.

[Editors' note: further revisions were suggested prior to acceptance, as described below.]

Thank you for submitting your article "*Drosophila* STING protein has a role in lipid metabolism" for consideration by *eLife*. Your article has been reviewed by 3 peer reviewers, one of whom is a member of our Board of Reviewing Editors, and the evaluation has been overseen by K VijayRaghavan as the Senior Editor. The reviewers have opted to remain anonymous.

Essential revisions:

1) The statistics have been corrected and the control genetic background seems to have been addressed, although the outcrossing is still a bit vague in the methods. Have all chromosomes in the background of each genotype been replaced with *w^1118^* wild-type chromosomes, or are there floating CyO or other mutant chromosomes present that differ across genotypes?

2) Some graphs still lack Y-axis labels (Figure 5B for example) and only some graphs have been edited to show the data points as requested. It would be best if all graphs had the same formatting throughout the paper.

3) Also in Figure 5B, for ATP, the *** comparison bar is shifted to the right.

4) For the adult expression data, Figure 1 supplement 1 panel B, I found it surprising that they did not measure expression in the fat body, which is a key tissue of interest. Furthermore, they say that the adult carcass consists primarily of cuticle and fat body even though there is quite a bit of muscle there. The fat body isn't typically associated with the carcass and dissociates easily. So, an explicit description of how these tissues were isolated and quantified would be helpful. The methods don't seem to cover the Westerns done for this figure.

5) Figure supplement 1 contains data the authors didn't generate (from modENCODE) that may not be appropriate to publish as primary data from the authors here. I'm not sure.

6) Figure 8 should say lipid biosynthesis instead of anabolism, which is an outdated term. There also needs to be a more detailed legend for Figure 8.

7) The figures should be labelled as Figure 1 or Figure 1 Supplemental Figure 1.

In the current version, it is hard to track which one is what (I had to go back and forth to match them).

8) The lipidTOX label is missing from the panel.

*Reviewer #1:*

Most of the essential revisions have been completed satisfactorily.

*Reviewer #2:*

This is a much-revised version of the earlier manuscript.

The claims are now supported to a large extent.

The authors have addressed most of the comments raised in the previous round.

*Reviewer #3:*

The authors addressed almost all of the reviewer concerns very well and I think the manuscript is close to being ready for publication.

The statistics have been corrected and the control genetic background seems to have been addressed, although the outcrossing is still a bit vague in the methods. Have all chromosomes in the background of each genotype been replaced with *w^1118^* wild-type chromosomes, or are there floating CyO or other mutant chromosomes present that differ across genotypes?

Some graphs still lack Y-axis labels (Figure 5B for example) and only some graphs have been edited to show the data points as requested. It would be best if all graphs had the same formatting throughout the paper.

Also in Figure 5B, for ATP, the *** comparison bar is shifted to the right.

For the adult expression data, Figure 1 supplement 1 panel B, I found it surprising that they did not measure expression in the fat body, which is a key tissue of interest.

Furthermore, they say that the adult carcass consists primarily of cuticle and fat body even though there is quite a bit of muscle there. The fat body isn't typically associated with the carcass and dissociates easily. So, an explicit description of how these tissues were isolated and quantified would be helpful. The methods don't seem to cover the Westerns done for this figure.

Figure supplement 1 contains data the authors didn't generate (from modENCODE) that may not be appropriate to publish as primary data from the authors here. I'm not sure.

Figure 8 should say lipid biosynthesis instead of anabolism, which is an outdated term. There also needs to be a more detailed legend for Figure 8.

---

## [Author Response]

Essential revisions:1. A central message of this study is a direct role of dSTING in regulating metabolism, independent of its role in immunity. The potential chronic infection status of the flies in the experiments presented contributing to the metabolic changes observed is an important concern – reduced immunity in dSTING mutants could produce a higher pathogen load, which would consume energy. Please perform the storage metabolite assays (TAG, glycogen) under axenic conditions would help show that the two are or are not linked.

The data on starvation stress and oxidative stress of axenic flies are included and supplemented with genetic rescue data (Figure 1—figure supplement 3A,B). TAG and glycogen levels under axenic conditions are measured and presented in Figure 1—figure supplement 3C,D. The data indicate that *dSTING* mutant flies show decreased TAG and glycogen levels compared to the control flies under axenic conditions.

2. Cg-Gal4 expresses in fat body/circulation as well as lymph gland hemocytes. yolk-gal4 (adult-specific fat body driver) should be used to exclude the contribution of hemocytes. This is mandatory to exclude the contribution of non-fat body cells towards the phenotype.

To address this concern we performed RNAi of *dSTING* using adult female fat body-specific *yolk-GAL4* driver. We confirm that female flies with reduced dSTING expression specifically in fat body are more susceptible to starvation stress and oxidative stress and have diminished TAG and glycogen levels. These data are presented in Figure 2—figure supplement 3.

3. In the manuscript experiments on both adult and larval fat body are shown in various figures. Reasons for the stages used and timeline of the experiments need to be stated clearly. The larval and adult stages cannot be used interchangeably. Assuming that the larval and adult fat body are equivalent is not advisable. Uniformity of the experimental set up is mandatory.

The manuscript is rewritten and reorganized to address this concern. Some of the experiments regarding larval fat body were removed from the manuscript to avoid confusion such as autophagy data (previous Supplementary Figure 5) and lipid staining of larval fat body (previous Supplementary Figure 2C). We also added the description of the age and sex of the flies used in the experiments in Figure legends.

4. Spatiotemporal analysis of STRING across tissues is needed to further endorse/support the function proposed.

We analyzed the expression pattern of *GFP-dSTING* across adult and larval tissues. The results are included in the Figure 1—figure supplement 1A. The paragraph describing these results was added to the text. The highest level of expression was observed in the digestive tract in both adults and larvae. *GFP-dSTING* was also expressed at the high level in larval fat body and adult carcasses (which mainly consist of cuticle and fat body cells) (Figure 1—figure supplement 1B,D). Our results are consistent with the modENCODE Tissue Expression Data for *dSTING* (Brown et al., 2014) (Figure 1—figure supplement 1C,E).

5. Among the changes in various metabolites presented, only that in TAG is convincing; others such as trehalose are less compelling. When comparing fed v starved in any genotype or between two genotypes, it would be best to show the changes in levels of a metabolite as a fractional/proportional change and then compare changes in these between genotypes.

We agree that the phenotype for TAG is the most convincing. These particular results drove our interest to look closer to the lipid metabolism in mutant flies and it became a main focus of the manuscript.

6. Technical issues that must be addressed(i) All survival curves are discontinuous data and authors should not present them as continuous data as in all the figures of the current version of the manuscript.

We changed survival curves as per your suggestion.

(ii) genetic and biochemical (IP, mass spec) controls need to be explicit for all experiments, including genetic background controls, and measuring the degree of expression when overexpressing and after knockdown.

We added all the controls to all Figure legends. We measured the expression level of *dSTING* in mutants and in genetic rescue flies (Figure 1—figure supplement 1A).

(iii) A genetic rescue is missing in many cases, needs to be done to add robustness to the hypothesis. Genetic rescue of at least one larval phenotype with the GFP-STING is required.

The genetic rescue data are added to the following figures: Figure 1—figure supplement 2 (Larvae, starvation and oxidative stress), Figure 1—figure supplement 3 (axenic flies, starvation and oxidative stress, TAG and glycogen levels).

(iv) The expression levels of dSTING-GFP in genotypes where it is used to rescue mutant phenotypes must be presented.

We measured the expression level of *GFP-dSTING* in genetic rescue flies. The data are presented in (Figure 1—figure supplement 1A). The level of *dSTING* expression in *dSTINGΔ;GFP-dSTING* flies was the same as in control flies.

(v) The effectiveness of the RNAi line used in this study needs to be shown experimentally by RT-PCR.

RNAi of *dSTING* in adult fat body using yolk-GAL4 driver resulted in approximately two-fold reduction in *dSTING* expression as measured by qRT-PCR. These data are presented in Figure 2—figure supplement 3D.

Reviewer #1:[…] Comments for the authors:1. What is the expression pattern of STING across tissues and developmental times in *Drosophila*? Does this support the function proposed here ?

We analyzed the expression pattern of *GFP-dSTING* across adult and larval tissues. The results are included in the Figure 1—figure supplement 1. The paragraph describing the results was added to the text. The highest level of expression was observed in the digestive tract in both adults and larvae. *GFP-dSTING* was also expressed at the high level in larval fat body, a major lipid-synthesizing tissue in flies, and adult carcasses (which mainly consist of cuticle and fat body cells (Figure 1—figure supplement 1B,D)). Our results are consistent with the modENCODE Tissue Expression Data for *dSTING* (Brown et al., 2014) (Figure 1—figure supplement 1C,E).

2. All survival curves are discontinuous data and authors should not present them as continuous data as in all the figures of the current version of the manuscript.

The presentation of the survival curves is changed as per your suggestion.

3. Among the changes in various metabolites presented, only that in TAG is convincing; others such as trehalose are less compelling. When comparing fed v starved in any genotype or between two genotypes, it would be best to show the changes in levels of a metabolite as a fractional/proportional change and then compare changes in these between genotypes.

We agree that the phenotype for TAG is the most convincing. These particular results drove our interest to look closer to the lipid metabolism in mutant flies and it became a main focus of the manuscript.

As for the representation of data, we think that absolute values (μg metabolite per μg protein) are more informative than the proportional changes. In addition, as suggested by Reviewer 3 (point 2), we changed the bar graphs to scatter plots throughout the manuscript.

4. "Together, our results demonstrate a direct role of STING in lipid metabolism in *Drosophila*."This statement in the abstract is too strong given the data available in the manuscript.

We corrected this statement in the abstract.

5. Trehalose is synthesized in the fat body from glucose. Given the noted reduction in trehalose levels and elevation in glucose levels, is trehalose synthesis affected in STING mutants ?

Trehalose is synthesized by the trehalose synthesis enzyme Tps1 and is hydrolyzed by trehalase Treh. We measured the expression levels of both enzymes with RT-qPCR and found no difference in *dSTING* mutants compared to the wild type. However, since we used whole bodies we cannot exclude the possibility that there is a difference on tissue level. Also, there might be a difference on the protein level. We speculate that the reduced amount of storage TAG in mutant flies makes them feel “starved” resulting in glucose release from glycogen and trehalose.

6. GFP-dSTING construct: Please show the level of expression of this construct when expressed in mutant background relative to wild type sting levels.

We measured the expression level of *GFP-dSTING construct* in genetic rescue flies. The data are presented in Figure 1—figure supplement 1A. The level of *dSTING* expression in *dSTINGΔ;GFP-dSTING* flies was the same as in control flies.

7. Overexpression of STING had no effect given its proposed role as a direct regulator of metabolism. Related to the point above.

We cannot rule out the possibility that *dSTING* overexpression may have some effect on lipid metabolism and/or ACC and FASN interaction. The only experiment we performed with flies overexpressing *dSTING* in fat body was the IP experiment for ACC and FASN interaction (Figure 4C). We did observe more FASN interacting with ACC in “rescue” or “overexpression” flies compared to control and mutant flies, however we do not consider IP as a quantitative method. More experiments will be needed to make the conclusion.

8. (Pospisilik et al., 2010) (Baumbach et al., 2014) "whereas in our study dSTING was absent from the very beginning of the development. "This indicates a developmental role for STING which is unclear.

We agree that *dSTING* might have a role in development. We observed slightly reduced rate of pupation in *dSTING* mutants (Figure 1—figure supplement 2B). More experiments will be needed to address the role of dSTING in development.

9. "Together, these data indicate that a deletion of dSTING results in a disruption of lipid/carbohydrate energy balance." This statement does not appear substantiated.

We removed this statement from the manuscript.

10. An important point to note is that the GAL4 driver used here expresses in fat body, hemocytes and lymph gland. This needs to be addressed since we are talking about a molecule that already has a demonstrated role in immune function. Otherwise, it will be difficult to uncouple the immune and metabolic role of STING.

We performed RNAi of *dSTING* using adult female fat body-specific *yolk-GAL4* driver. Female flies with reduced dSTING expression specifically in fat body were more susceptible to starvation stress and oxidative stress and had lowered TAG and glycogen levels. These data are presented in Figure 2—figure supplement 3.

11. The effectiveness of the RNAi line used in this study needs to be shown experimentally by RT-PCR.

RNAi of *dSTING* in adult fat body using *yolk-GAL4* driver resulted in approximately two-fold reduction in *dSTING* expression as measured by qRT-PCR. The data are presented in Figure 2—figure supplement 3D.

12. The role of dSTING independent of the INR/Foxo pathway, as presented here is not at all convincing. See also details on specific figures below.

We removed the data on Foxo pathway from the manuscript to improve readability of the text and to reduce deviation from the main line of the manuscript.

13. The IP and mass spec on dSTING. What are the controls and what did you do to ensure that the interactions seen are likely to be specific?

For IP we used larvae ubiquitously overexpressing *GFP-dSTING* (genotype *w^1118^;+/+;tub-GAL4/GFP-dSTING*). *w^1118^* larvae were used as a control. To ensure that the interactions are specific, recombinant GFP protein was added to the control extract. Also, high detergent was used in the IP buffer (1% NP-40) and beads were pre-cleaned for 4 hours. IP was performed using anti-GFP antibody. All the information is provided in Materials and methods section.

14. Many times, the paper switches between adults and larvae for expts and also uses adults and larvae at various points interchangeably. For e.g. mas spec done from larval FB and confirmatory IP from adult abdomen. This is not advisable as the phenotypes presented here likely have a developmental origin. IT cannot also be assumed that the larval and adult fat body are equivalent.

The manuscript is rewritten and reorganized to address this concern. Some of the experiments regarding larval fat body were removed from the manuscript to avoid confusion such as autophagy data (previous Supplementary Figure 5) and lipid staining of larval fat body (previous Supplementary Figure 2C) as suggested. We also added the description of the age and sex of the flies used in the experiments in Figure legends.

As for the mass spec, we performed the mass spec experiment on larvae as one of our initial experiments, since larval fat body is very easy to dissect relative to the adult fat body. Most prominent hits were then confirmed using IP from adult carcasses extracts (Figure 4).

15. Supp Figure 7. Many large changes in metabolomics data left unexplained and not discussed.

The metabolomics data are provided for the information only. We feel that the discussion of specific changes in a multitude of different metabolites would deviate from the main focus of the manuscript and require a separate study.

16. Given the already well-known role of STING in mammals the authors should present data on:(i) Any changes in mitochondria in dSTING mutants?

We performed immunostaining of mitochondria and did not see the difference in *dSTING* mutants compared to the control. However, we did not measure the activity of mitochondria. These experiments will require further investigation.

(ii) Any evidence for DNA damage in dSTING mutants?

We did not observe any evidences of DNA damage in *dSTING* mutants under normal conditions. The analysis of the DNA damage in *dSTING* mutants under stress will require additional further studies outside of the scope of current manuscript.

(iii) Strengthen the data that the metabolic changes seen in dSTING are not an indirect outcome of ongoing infection. Current one data is not strong enough.

We measured TAG and glycogen levels in axenic flies and observed diminished levels of these metabolites in mutants as compared to the control flies (Figure 1—figure supplement 3C,D). Similarly, TAG and glycogen levels were decreased in flies with fat-body specific RNAi of *dSTING* (Figure 2—figure supplement 3C,D).

17. Figure 2E' and 2F'. Quantification not clear. What do you mean by area of lipid droplets? Not related to conclusion claimed.

We substituted “area of lipid droplets” with “Surface area occupied by lipid droplets” to avoid confusion. To calculate it we randomly chose equal squares on Nile-red labelled fluorescent images and calculated total area covered by Nile-red staining at each square.

18. Suppl Figure 1C – quantification needed.Suppl Figure 3 A. not clear what is 24 hrs and 48 hrs.Suppl Figure 3B not clear.

We corrected specified Figures and Figure legends.

19. Suppl Figure 6. The fractionation of cells and the purity of the fractions not shown. Controls needed here. Fraction markers as in Figure S8 needed. Data presented here is not convincing. Positive control also needed.

We removed the data on Foxo pathway from the manuscript to improve the narrative and readability of the text as it does not add much to the main focus of the manuscript (lipid metabolism).

Reviewer #2:Akehmetova et al. intend to demonstrate an unappreciated role of dSTING in lipid metabolism. Although dSTING deleted flies show no error in metabolism, they are sensitive to starvation and oxidative stress. The authors discovered that upon starvation/ oxidative stress, there is a significant decrease in the storage metabolites, such as TAG, trehalose and glycogen in dSTING deleted flies.They identified two essential fatty-acid biosynthesis enzymes – Acetyl-CoA carboxylase (ACC) and fatty acid synthase (FAS) – as interacting partners through molecular and biochemical approaches for dSTING.They propose that dSTING, FAS and ACC interacted with each other, and probably all three proteins are components of a large complex. In the absence of dSTING, there is a decrease in FAS activity and defects in ACC cellular localization, suggesting a direct role of dSTING in the lipid metabolism of fruit flies.These data add an interesting insight into our understanding of stress-induced altered metabolism. Data mostly well support this paper's conclusions, but some aspects of image acquisition and data analysis need to be clarified and extended. Following points should be addressed.1) Title: "*Drosophila* STING protein has a role in lipid metabolism". The title is misleading as a total loss of STING does not have a phenotype. The phenotypes are detectable upon stress. It emphasizes the fact that the dSTING function enables the organism to tide over stress. This thought should be reflected in the title.

We think that the title accurately represents our findings. We also would like to point out that *dSTING* mutant flies have phenotype of decreased TAG, glycogen and trehalose levels even under fed conditions.

2) The timeline for starvation/ oxidative stress should be schematically present or mentioned at least once in the text.

We added following sentences to the manuscript: “After the eclosion, flies were kept on regular food for 5 days and then transferred to vials containing wet Whatman paper (starvation stress) or to vials containing regular food supplemented with 5% hydrogen peroxide (oxidative stress)”. Also, we added the description of the age and sex of the flies used in the experiments in Figure legends.

3) the dSTING starved larvae do they have a delay in pupation? Do the pupae eclose? Since pupae is a non-feeding state, the effect that we see in larvae will be nice to extend in the pupal stage.

We have preliminary data on slightly reduced pupation rate for *dSTING* mutants even in fed conditions (Figure 1—figure supplement 2B). However, in this manuscript we concentrate on adult stage. Future experiments will be needed to explore the effect of *dSTING* mutation at larval and pupae stages.

4) "The expression of GFP-dSTING partially or entirely rescued the sensitivity of dSTINGΔ deletion flies to both starvation and oxidative stress (Figure 1B,C), suggesting that observed phenotypes are caused by dSTING deficiency. " Can it rescue larval defects…it needs to be shown.

We added genetic rescue data to larval starvation and oxidative stress tests (Figure 1—figure supplement 2AB).

5) "the age-related mortality was slightly reduced, especially for females (Figure 1D)". A possible explanation needs to be mentioned.

There are multiple evidences that the reduced TAG level correlates with the increased lifespan (PMID: 16434470, PMID: 21930912, PMID: 22870336, PMID: 12543978, PMID: 31560163). One possibility is that increased longevity is the direct result of altered insulin signaling. Mutations that reduce signaling through the insulin-like signaling pathway can increase lifespan in *Drosophila* (eg PMID: 11292874, PMID: 11292875). We did observe decreased expression of several DILPs using microarray and RT-qPCR data. However, more experiments will be needed to determine the exact mechanism.

6) "Two RNAi screens for obesity and anti-obesity genes in *Drosophila* did not reveal any significant changes in TAG level in dSTING-deficient flies (Pospisilik et al., 2010) (Baumbach et al., 2014). The potential discrepancy with our data might be explained by the fact that in both mentioned studies RNAi was induced only 2-8 days after eclosion, whereas in our study dSTING was absent from the very beginning of the development" – Question: was the starvation in the current study not limited to the adulthood? To avoid confusion, the regime of starvation, a scheme is needed. If RNAi is done from the very beginning, will it endorse the current findings?That might resolve the discrepancy.

In this manuscript we concentrated on the role of *dSTING* in mature 5-days-old adults. Shortly after the eclosion flies do not have a fat body yet, instead they rely on the fat cells that are a leftover from the larval fat body. Therefore we used 5-days old flies in our experiments to allow the adult fat body to form.

The description of starvation regimen was added to the manuscript. Also, we added the description of the age and sex of the flies used in the experiments in Figure legends.

7) Additional marker for neutral lipid should be used (LipidTOX).

We performed the staining of adult fat body using LipidTox, data are presented in Figure 2—figure supplement 2.

8) "fat body specific RNAi of dSTING resulted in starvation sensitivity and reduced TAG level in adult flies (Supplementary Figure 4), highlighting the role of dSTING in fat body functions." Cg-Gal4 expresses in also in blood/hemocytes. A more specific driver for the fat body like yolk-gal4 (adult-specific) should be used to exclude the contribution of hemocytes.

We performed RNAi of *dSTING* using adult female fat body-specific *yolk-GAL4* driver. The female flies with reduced dSTING expression specifically in fat body were more susceptible to starvation stress and oxidative stress and had lowered TAG and glycogen levels. These data are presented in Figure 2—figure supplement 3.

9) "Consequently, we did not observe the increase in nuclear FoxO which is a characteristic of FoxO activation (Supplementary Figure 6)". Upon starvation, both dSTING lof and control upregulated nuclear FoxOcompared to fed.…any comments?

We removed the data on Foxo pathway from the manuscript to improve readability of the text and to reduce deviation from the main focus of the manuscript.

10) (Figure 5C, Supplementary Figure 7) Please mention the loading control, not as pnut but peanut or cpr=NADPH-cytochrome P450 reductase in the figure legend or the panel itself.

We added the full protein names to the Figure legends.

11) "Both ACC and FAS partially co-localized with GFP-dSTING at the cell periphery region of the ER (Figure 6B,C)". The Calnexin labeling performed is not convincing.It needs to be repeated and a higher magnification to be included.

We included higher magnification in Figure 6. We think that the reason for an unusual Calnexin staining is that most of the fat body cell volume is occupied by lipid droplets. Our Calnexin labelling is in agreement with (Jacquemyn et al., 2020).

12) 7A: higher magnification to be included.

We included higher magnification in Figure 7B.

13) Strangely enough, how can ACC activity remain unchanged even though it is mislocalized upon loss of dSTING? ACC localization is not essentially related to its activity?14) The FAS protein is normal, but activity is decreased. Is it due to its partial colocalization with the aggregates of ACC in the absence of STRING? It needs to be discussed.

The result is also puzzling for us. We hypothesize that FASN activity is improved when in the complex with ACC (for example by a faster transfer of malonyl-CoA product of ACC to the active site of FASN). When ACC is mislocalized from FASN in *dSTING* mutants, ACC activity remains unchanged but FASN activity is decreased. We did observe less FASN immunoprecipitated with ACC in *dSTINGΔ* mutants compared to the control flies, and the opposite effect was found in flies expressing GFP-tagged dSTING (Figure 4C).

15) "Reliance on glucose for energy also has a consequence of reduced incorporation of glucose into trehalose and glycogen for storage, and therefore, lower levels of these storage metabolites, which we observed (Figure 2B,C)" – what happens if we inhibit glucose by 2DG or genetically in String loss: will there be an increase in the levels of glucose into trehalose and glycogen?

This is a very interesting idea, and we plan to test it. However for this manuscript we decided to concentrate more on the effect on lipid metabolism to improve the narrative and readability.

Reviewer #3:et al. *Drosophila*[…] There are some flaws in the experimental design and analyses that are of concern and prevent me from recommending acceptance.1. The genetic background of the control flies should match the mutant alleles being used. For the deletion, there should be a precise excision of the P element used to generate the deletion, so that the genetic background is matched between the two. The genetic background of this insertion is not w^1118^ but y[1] w[67c23]. For UAS crosses, the correct control should be a cross between the driver and an empty insertion site and/or the UAS crossed into the driver background. It is not clear to me if this was used consistently. The w^1118^ stock control used throughout the paper is not the correct control and therefore it is difficult to interpret your results.

During the crosses to obtain *dSTING* deletion, the chromosome with the excision was transferred to *w^1118^* background. Therefore, our control, mutant and rescue flies all have same *w^1118^* background. Also, we measured TAG levels in flies *w^1118^;orc6^35^/orc6^35^;FLAG-Orc6WT*. In this stock both *Orc*6 and *dSTIN*G genes are deleted due to the *orc6^35^* excision (PMID: 19541634), Orc6 deletion is rescued by *FLAG-Orc6WT* expression, but *dSTING* mutation is not rescued. These flies also have reduced TAG levels similar to *dSTINGΔ* mutants.

**Author response image 1. sa2fig1:** 

2. For some experiments, there seem to be low replicate #s (3-5) and the bar graphs showing the means instead of individual data points make it difficult to determine whether the data are normally distributed. Scatter plots with each data point would help the reviewers better assess the rigor of these experiments.

We changed bar graphs to scatter plot graphs throughout the manuscript.

3. The survival analyses in 1B, C, D and elsewhere should use Mantel-Cox statistics, not t tests. And comparisons of more than two sample types should probably use an ANOVA, not a student's t-test.

We changed student’s t-test to ANOVA test where necessary, and performed log-rank test for survival curves.

4. It would also be great if quantitative data were presented for the Western blots. A couple of the blots seem unevenly loaded (5B', S6 fed) and in Figure S6, a nuclear protein should have been used as a loading control for nuclear fractions.

We added quantifications for Figure 5A’,5B’. We removed the data on Foxo pathway (including Figure S6) from the manuscript to improve readability of the text and to reduce deviation from the main focus of the manuscript.

[Editors' note: further revisions were suggested prior to acceptance, as described below.]

Essential revisions:1) The statistics have been corrected and the control genetic background seems to have been addressed, although the outcrossing is still a bit vague in the methods. Have all chromosomes in the background of each genotype been replaced with w^1118^ wild-type chromosomes, or are there floating CyO or other mutant chromosomes present that differ across genotypes?

All chromosomes in the background of each genotype were replaced with *w^1118^* wild type chromosomes. Since during the experiments we used flies homozygous for *dSTING* mutation as well as flies homozygous for both *dSTING* mutation and *GFP-dSTING* transgene, we are confident that there is no floating *CyO* or other mutant chromosomes.

2) Some graphs still lack Y-axis labels (Figure 5B for example) and only some graphs have been edited to show the data points as requested. It would be best if all graphs had the same formatting throughout the paper.

We added Y-labels to all graphs.

Graph in Figure 1—figure supplement 2B (oxidative stress in larvae) was changed to show each data point.

Some of the graphs were left unchanged since they show values relative to the control (e.g. metabolite levels in Figure 5—figure supplement 1B or RT-qPCR data in Figure 1—figure supplement 1A).

3) Also in Figure 5B, for ATP, the *** comparison bar is shifted to the right.

We corrected the error bar.

4) For the adult expression data, Figure 1 supplement 1 panel B, I found it surprising that they did not measure expression in the fat body, which is a key tissue of interest. Furthermore, they say that the adult carcass consists primarily of cuticle and fat body even though there is quite a bit of muscle there. The fat body isn't typically associated with the carcass and dissociates easily. So, an explicit description of how these tissues were isolated and quantified would be helpful. The methods don't seem to cover the Westerns done for this figure.

We apologize for the confusion. We changed “carcass” to “abdominal carcass” throughout the article. Adult abdominal carcass is what's left of the abdomen after the gut and testes/ovaries have been removed. In the literature abdominal carcasses are considered to be enriched in fat body (PMID: 28669758, PMID: 31080057, PMID: 28704946, PMID: 20689503). When we dissect adult abdominal carcass we leave the fat body attached to the body wall and we have never observed that it dissociates from it.

The paragraph named “GFP-dSTING tissue expression” was added to the Material and methods section.

5) Figure supplement 1 contains data the authors didn't generate (from modENCODE) that may not be appropriate to publish as primary data from the authors here. I'm not sure.

We would like to include these data since we use them to compare with our results. In the figure legend we clarify that these are the modENCODE tissue expression data and make a reference (Brown et al., 2014).

6) Figure 8 should say lipid biosynthesis instead of anabolism, which is an outdated term. There also needs to be a more detailed legend for Figure 8.

We changed Figure 8 according to your suggestion and expanded Figure legend.

7) The figures should be labelled as Figure 1 or Figure 1 Supplemental Figure 1.In the current version, it is hard to track which one is what (I had to go back and forth to match them).

The figures in the manuscript are labeled as per *eLife* submission rules. The related manuscript file contains Figures with labels and legends.

8) The lipidTOX label is missing from the panel.

The data on LipidTox staining are presented in Figure 2—figure supplement 2 as indicated in the Figure legend and in the text.